# Iron absorption and loss, and efficacy of iron supplementation with and without prebiotics in children with virally suppressed HIV: three prospective studies in South Africa

Children living with HIV are at risk for iron deficiency, yet optimal strategies for prevention and treatment remain unclear. Here, we investigate iron absorption, losses, and the efficacy and safety of oral iron supplementation with versus without prebiotics in three prospective studies in children with virally suppressed HIV in South Africa (NCT03572010, NCT04931641). In the first study, using stable iron isotopes, we show that iron absorption from iron-fortified maize porridge, a lipid-based nutrient supplement, and an oral iron supplement is comparable between children with HIV ($n = 43$) and without HIV ($n = 45$). In the second study, we use a stable iron isotope dilution method over a 6-month period to demonstrate that children with HIV ($n = 29$) absorb significantly less iron from their habitual diet than their uninfected peers ($n = 36$), while basal iron losses are similar. In the third study, a 12-week randomised, placebo-controlled, double-blind trial, iron-deficient children with HIV receiving iron with prebiotic galacto-oligosaccharides ($n = 41$) exhibit a 39% greater relative increase in serum ferritin (primary outcome) compared to those receiving iron with placebo ($n = 42$) ($p = 0.053$). They also report significantly fewer infection-related symptoms, with no significant differences in gut inflammation or enteropathogen carriage (secondary outcomes). Collectively, these findings indicate that while dietary iron absorption is reduced in children with virally suppressed HIV, supplemental and fortificant iron are well absorbed, and co-administration of iron supplements with prebiotics may improve efficacy and safety.

In 2024, an estimated 1.4 million children aged 0–14 years were living with HIV worldwide[1]. Anaemia is a common complication of paediatric HIV[2] and is associated with poorer clinical outcomes and increased mortality[3]. The global prevalence of anaemia among children under 15 years with HIV is ~40%, with reported rates ranging from 26% to 61%[2], depending on setting and disease stage. The aetiology of anaemia in HIV is multifactorial, involving nutritional deficiencies, inflammation, and HIV-specific factors[3]. In adults living with HIV, iron deficiency accounts for an estimated 20–44% of anaemias, while inflammation contributes to 41–47%, with substantial overlap between these causes[3].

✉ e-mail: michael.zimmermann@rdm.ox.ac.uk

Current strategies to prevent and treat iron deficiency in children include dietary counselling to increase intake of iron-rich foods, food fortification, and oral iron supplementation[4], including the use of iron-containing lipid-based nutritional supplements (LNS). However, even among children with HIV receiving antiretroviral treatment (ART), low-grade systemic inflammation[5] and ongoing viral replication in the gut mucosa[6–8] may persist, potentially impairing iron absorption or increasing gastrointestinal iron losses. Despite these concerns, to our knowledge, iron absorption from fortified foods and supplements has not been previously measured in children with HIV using isotopic methods, nor directly compared to children without HIV.

Previous studies investigating the response to iron interventions in children with HIV have relied on conventional iron status biomarkers[9,10] such as serum ferritin and soluble transferrin receptor (sTfR), both of which are influenced by inflammation and may not reliably reflect iron status[3,11]. In contrast, stable isotope techniques provide a direct, accurate and safe method for quantifying iron absorption and losses, independent of inflammation[12]. A recently developed isotope dilution method enables precise quantification of long-term dietary iron absorption and endogenous losses in African children[13], and may be particularly valuable for studying iron kinetics in children with HIV. Furthermore, prebiotic fibres such as galacto-oligosaccharides (GOS) can modulate gut immune function and enhance iron absorption[14], offering a potential strategy to improve the efficacy and safety of oral iron supplementation in iron-deficient children living with HIV.

To address key knowledge gaps and explore new approaches to improve iron interventions in this population, we conducted three complementary prospective studies in South Africa. Each study focused on distinct but related aspects of iron metabolism in children with virally suppressed HIV:

1. Study 1 compared iron absorption from a single serving of iron-fortified maize porridge, LNS, and an oral iron supplement between children with and without HIV using stable iron isotopes.
2. Study 2 quantified habitual dietary iron absorption and iron losses over 6 months using the isotope dilution method in children living with and without HIV.
3. Study 3 evaluated the efficacy and safety of oral iron administered with versus without GOS in iron-deficient children with virally suppressed HIV in a randomised, double-blind, placebo-controlled trial.

We hypothesized that: 1) iron absorption from common iron interventions would be lower in children with HIV compared to those without; 2) habitual dietary iron absorption would be lower, and iron losses higher, in children with HIV; and 3) in iron-deficient children with HIV, co-administration of GOS with oral iron supplements would result in greater increases in serum ferritin concentrations (biomarker of iron stores), and lower gut inflammation and adverse side effects, than iron alone.

In this work, we show that although children with virally suppressed HIV had higher systemic inflammation and absorbed less iron from their habitual diet (Study 2), iron absorption from a single serving of fortified maize porridge, LNS, and an oral iron supplement was comparable to that of children without HIV (Study 1). In the randomised trial (Study 3), co-administration of GOS with oral iron in iron-deficient children with HIV resulted in greater increases in ferritin concentrations and reduced incidence of fever and rhinorrhea, without increasing gut inflammation or numbers of enteropathogens. These findings suggest that GOS may improve both the efficacy and safety of oral iron supplementation in iron-deficient children with virally suppressed HIV. Together, these studies comprehensively characterise iron absorption, losses, and response to current iron interventions in children with virally suppressed HIV. Improved understanding of iron metabolism and treatment response in this vulnerable group is essential to inform evidence-based strategies for the prevention and management of iron deficiency in paediatric HIV.

## Results

### Study 1: iron absorption from common iron interventions

Between September 2018 and September 2019, we conducted a prospective, cross-over study with case-control comparisons to assess fractional iron absorption (FIA; %) from three commonly used iron interventions, at doses representative of public health and clinical practice[15–17], in iron-depleted children with and without HIV. We screened 293 children aged 8–13 years and enrolled 90 iron-depleted children with virally suppressed HIV ($n = 45$) and without HIV ($n = 45$), matched for age and sex (assigned at birth [from health record]). All children completed the study, but two with HIV were excluded from analysis due to detectable viral loads (HIV RNA ≥ 50 copies/ml), leaving 88 children in the final analysis (Fig. 1).

Each child received three single-dose iron interventions on non-consecutive days under fasting conditions: (1) iron-fortified maize porridge labelled with 2 mg of $^{58}$Fe as ferrous fumarate, (2) a LNS labelled with 6 mg of $^{57}$Fe as ferrous sulphate, and (3) an oral iron supplement (55 mg elemental iron) labelled with 6 mg of $^{57}$Fe as ferrous sulphate. The order of the first two interventions was randomized. Blood samples were collected 14 days after administration of the first two interventions and again 14 days after the oral iron supplement to determine FIA based on erythrocyte incorporation of the stable isotopes. This method allows direct quantification of iron absorption from each intervention.

At baseline, the children with HIV had lower height-for-age Z scores ($P = 0.002$), higher sTfR concentrations ($P = 0.029$), lower hepcidin levels ($P = 0.001$), and more systemic inflammation ($P = 0.006$) than children without HIV (Table 1). There were also trends toward higher prevalence of stunting ($P = 0.054$), lower haemoglobin ($P = 0.068$), higher prevalence of anaemia ($P = 0.068$), and higher plasma intestinal fatty acid binding protein (IFABP) ($p = 0.059$), a biomarker of enterocyte damage, in the children with versus without HIV. Socio-economic and demographic characteristics are shown in Supplementary Table 1. Notably, breadwinners in households of children with HIV were less likely to be permanently employed (P < 0.042).

FIA (%) and total iron absorption (mg) from the iron-fortified maize porridge ($n = 83$), LNS ($n = 79$), and iron supplement ($n = 78$) in children with and without HIV are shown in Fig. 2. Two children with HIV were excluded from the final analysis because of detectable viral loads. Three children with HIV and one child without HIV did not consume the entire LNS portion. We could not collect sufficient blood during venepuncture from two children with HIV and three children without HIV on both days 17 and 31, as well as from one child with HIV on day 31. These children were excluded from the analyses. Among the children with and without HIV, median (IQR) FIA from the fortified maize porridge was 6.5% (3.8–10.9) and 7.1% (5.6–10.1) ($P = 0.256$), from the LNS was 5.4% (3.7–8.8) and 5.1% (3.5–7.4) ($P = 0.573$), and from the iron supplement was 22.0% (16.0–28.6) and 19.6% (13.9–24.0) ($P = 0.264$), respectively.

Results from linear regression models predicting FIA and hepcidin concentrations are presented in Supplementary Tables 2 and 3. In models without and with the HIV variable, age was a positive predictor of FIA from the maize porridge (B = 0.85, 95%CI 0.19, 1.51 and B = 0.86, 95%CI 0.19, 1.53, respectively) and the LNS (B = 1.07, 95%CI 0.51, 1.64 and B = 1.05, 95%CI 0.49, 1.62, respectively). sTfR, which increases when iron availability to tissues is low, was a positive predictor of FIA from the ferrous sulphate supplement (B = 0.73, 95%CI 0.02, 1.45 and B = 0.76, 95%CI 0.03, 1.50). The addition of the HIV variable did not increase the adjusted $R^2$. In models without and with the HIV variable, sTfR (B = −0.54, 95%CI −1.01, −0.07 and B = −0.49, 95%CI −0.97, −0.02, respectively) and IL-6 (B = 1.42, 95%CI 0.47, 2.38 and B = 1.37, 95%CI 0.42, 2.33, respectively) were negative and positive predictors of

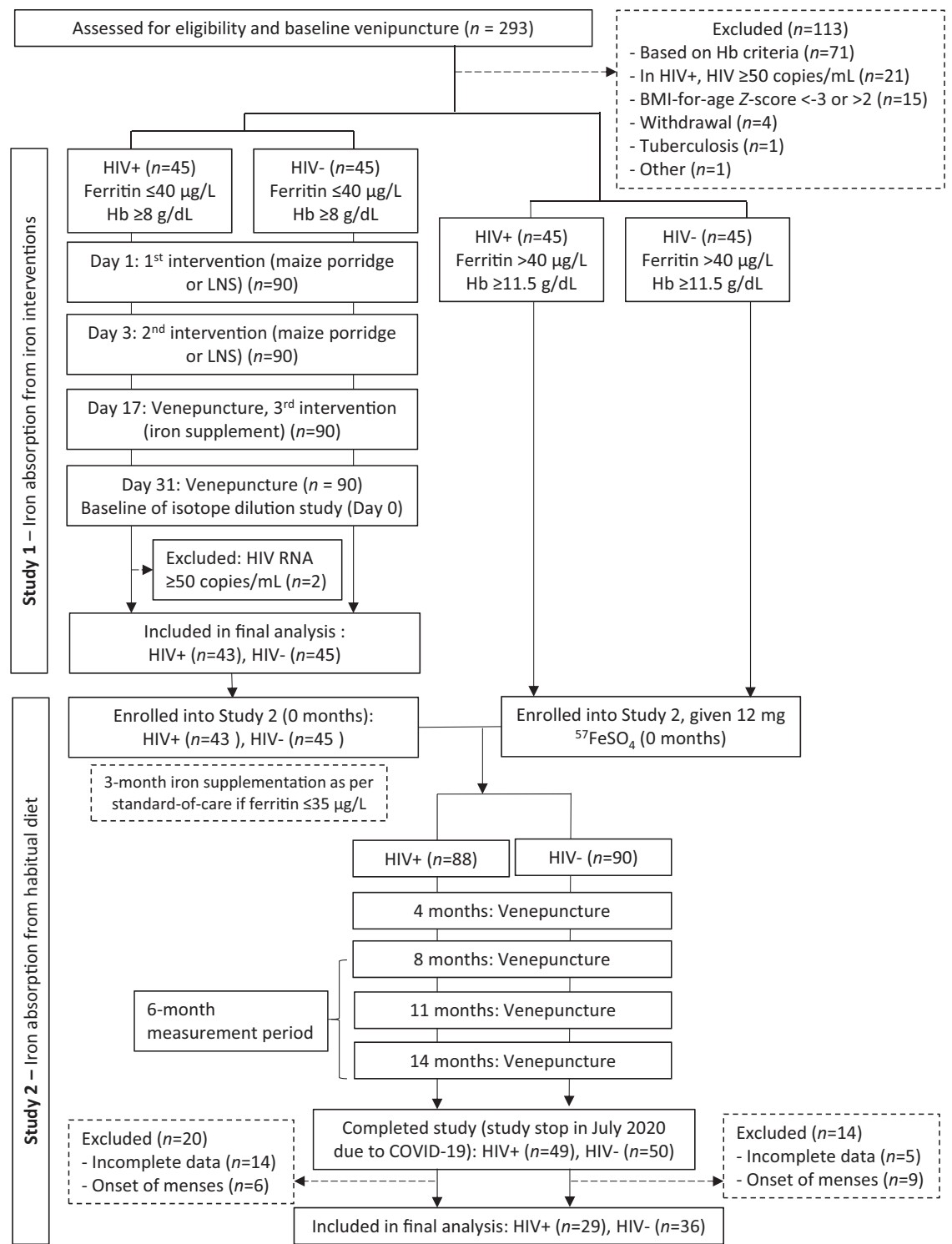

**Fig. 1 | Design and participant flow diagram for Studies 1 and 2.** Haemoglobin, haemoglobin; LNS, lipid-based nutrient supplement; FeSO₄, ferrous sulphate.

hepcidin, respectively, and the addition of the HIV variable did not increase the adjusted R².

## Study 2: iron absorption from the habitual diet and losses

Betweeen September 2018 and July 2020, we conducted a long-term isotope dilution study to quantify habitual dietary iron absorption (mg/day), iron losses (mg/day), and net iron balance (mg/day) over 6 months in children with and without HIV. A total of 180 children (with HIV: $n = 90$; without HIV: $n = 90$), aged 8–13 years, were labelled with 12 mg of $^{57}$Fe as ferrous sulphate, either during Study 1 or via a single

oral dose under fasting conditions. Following an 8-month equilibration period—during which the isotope distributed uniformly throughout body iron compartments—we followed the participants for an additional 6 months to measure dilution of isotopically enriched body iron by dietary iron of natural isotopic composition. This approach enabled precise quantification of daily iron absorption, iron losses, and net iron gains[12].

Due to the COVID-19 pandemic, the study was halted in July 2020 after 49 children with and 50 children without HIV, matched for age and sex, had reached the study endpoint. Of these, data from 34

**Table 1 | Study 1: Baseline characteristics of the iron-depleted children living with virally suppressed HIV and without HIV**

| | Children with HIV | Children without HIV | P-value |
|---|---|---|---|
| n | 43 | 45 | |
| Female/male sex assigned at birth, n (%) | 23/20 (54/46) | 23/22 (51/49) | 0.823 |
| Age, y | 11.5 (9.8–12.4) | 11.0 (9.6–12.2) | 0.512 |
| HIV RNA, copies/mL | <50 | - | - |
| Age at antiretroviral start, y | 1 (0–2.5) | - | - |
| Body-mass-index-for-age Z score | −0.39 ± 1.02 | −0.26 ± 1.09 | 0.570 |
| Underweight, n (%)[a] | 1 (2) | 4 (9) | 0.361 |
| Overweight, n (%)[a] | 4 (9) | 6 (13) | 0.739 |
| Height-for-age Z score | −1.38 ± 1.03 | −0.68 ± 1.01 | 0.002 |
| Stunted, n (%)[a] | 13 (30) | 6 (13) | 0.054 |
| Haemoglobin, g/dL | 12.0 ± 1.0 | 12.5 ± 1.1 | 0.068 |
| Anaemia, n (%)[b] | 15 (35) | 8 (18) | 0.068 |
| Ferritin, µg/L[c] | 16.5 (13.2-24.5) | 19.6 (14.4-25.5) | 0.415 |
| <15 µg/L, n (%) | 14 (33) | 12 (27) | 0.545 |
| Soluble transferrin receptor, mg/L | 7.0 (5.7-8.2) | 6.4 (5.4-7.4) | 0.029 |
| >8.3 mg/L, n (%) | 9 (21) | 5 (11) | 0.208 |
| Iron deficiency, n (%)[d] | 19 (44) | 15 (33) | 0.300 |
| Iron deficiency anaemia[e], n (%) | 10 (23) | 5 (11) | 0.130 |
| Transferrin saturation, % | 18.0 ± 9.5 | 19.8 ± 7.6 | 0.353 |
| Serum iron, µg/mL | 0.55 (0.38–0.83) | 0.58 (0.48–0.74) | 0.357 |
| Total iron binding capacity, µg/mL | 3.29 (2.93–3.61) | 3.24 (3.05–3.42) | 0.741 |
| C-reactive protein, mg/L | 0.13 (0.02–0.75) | 0.03 (0.01–0.05) | 0.009 |
| >5 mg/L, n (%) | 1 (2) | 0 (0) | 0.489 |
| α–1-glycoprotein, g/L | 0.54 (0.46–0.75) | 0.46 (0.39–0.61) | 0.021 |
| >1.0 g/L, n (%) | 5 (12) | 0 (0) | 0.025 |
| Inflammation, n (%)[f] | 6 (14) | 0 (0) | 0.006 |
| Interleukin-6, pg/mL | 1.1 (0.8–1.6) | 1.3 (0.9–1.9) | 0.333 |
| Hepcidin, ng/mL | 2.8 (1.8–5.2) | 5.1 (3.5–8.0) | 0.001 |
| Intestinal fatty acid binding protein, ng/mL[g] | 0.9 (0.5–1.2) | 0.6 (0.5–0.9) | 0.059 |
| Faecal calprotectin, µg/g[g] | 22.5 (11.5–65.2) | 26 (12.5–76.5) | 0.669 |
| Faecal pH[g] | 6.9 (6.4–7.2) | 7.1 (6.8–7.3) | 0.370 |

Values are medians (IQR), means ± SD, or n (%).

Independent-samples t-tests (two-sided) were used to compare continuous variables (non-normally distributed variables were log-transformed for analysis), and Pearson's Chi-squared tests (two-sided) or Fisher's exact tests (two-sided; when expected frequency of sample size was <5) were used to compare categorical variables between the two groups. No adjustments were made for multiple comparisons. Values are shown as either means ± SDs, or as medians (interquartile ranges).

[a]Stunting: height-for-age Z-score <-2; underweight: body-mass-index-for-age Z-score (BAZ) ≥−3 and <−2; overweight: BAZ > 1 and ≤2[9].

[b]Age <12 years: <11.5 g/dL; age ≥12 years: <12 g/dL[10].

[c]Adjusted for inflammation using BRINDA correction[30].

[d]Adjusted plasma ferritin <15.0 µg/L and/or soluble transferrin receptor >8.3 mg/L.

[e]Iron deficiency and anaemia.

[f]C-reactive protein >5.0 mg/L and/or α–1-glycoprotein >1.0 g/L.

[g]Assessed at Day 31 (study endpoint).

children were excluded from the final analyses (Fig. 1); 19 due to incomplete isotopic data (with HIV: $n = 14$; without HIV: $n = 5$) and 15 because girls began menstruating (with HIV: $n = 6$; without HIV: $n = 9$), to exclude bias from menstrual iron loss. The final analysis included 29 children with HIV and 36 without HIV, matched for age and sex. Of these, 21 previously participated in Study 1 (with HIV: $n = 9$; without

HIV: $n = 12$). Baseline characteristics are shown in Table 2. Compared to children without HIV, those with HIV had lower height-for-age ($P = 0.004$), weight-for-age ($P < 0.001$) and body mass index (BMI)-for-age Z-scores ($P = 0.003$). Socio-economic and demographic data are shown in Supplementary Table 4.

Over the 6-month measurement period, median (IQR) daily iron absorption ($Fe_{abs}$) was lower in children with HIV: 1.09 (0.82–1.32) mg/day versus 1.44 (0.85–2.06) mg/day in children without HIV ($P = 0.042$) (Fig. 3). This corresponds to a 24% lower relative iron absorption in children with HIV compared to their uninfected peers. Median (IQR) iron loss ($Fe_{loss}$) was similar between groups: 0.80 (0.43–1.13) mg/day in children with HIV and 0.78 (0.58–1.57) mg/day in those without HIV ($P = 0.059$). Net iron balance ($Fe_{gain}$) was also comparable: 0.33 (−0.03, 0.74) mg/day in children with HIV and 0.42 (0.04–0.90) mg/day in those without HIV ($P = 0.775$). In a sensitivity analysis that included the 15 menstruating girls, median (IQR) daily iron absorption ($Fe_{abs}$) was not significantly different between children with HIV ($n = 35$) and those without HIV ($n = 45$): 1.10 (0.88–1.47) mg/day vs. 1.42 (0.95–1.81) mg/day, respectively ($p = 0.119$). However, median (IQR) iron loss ($Fe_{loss}$) was significantly lower in children with HIV: 0.90 (0.46–1.16) mg/day compared to 0.97 (0.61–1.77) mg/day in children without HIV ($p = 0.036$). Net iron balance ($Fe_{gain}$) was similar between groups: 0.29 (−0.05 to 0.72) mg/day in children with HIV and 0.32 (−0.29 to 0.84) mg/day in those without HIV ($p = 0.372$).

There was a trend toward higher sTfR concentrations in children with than without HIV (Group, $P = 0.058$), and a significant time x group interaction for inflammation-adjusted ferritin concentrations ($P = 0.005$); only children without HIV showed an increase in ferritin over the 6-month period ($P = 0.043$). Systemic inflammation was more prevalent in children with HIV over the 6-month period (Group, $P < 0.001$). Dietary intake data are shown in Supplementary Table 5. Compared to children without HIV, those with HIV had lower median (IQR) intakes of haem iron (2.4 [1.6–3.6] versus 4.3 [3.2–5.9] mg/day; $P = 0.001$), animal protein (31 [26–40] versus 58 [42–72] g/day; $P < 0.001$), and vitamin C (51 [39–79] versus 93 [65–169] mg/day; $P = 0.002$). In contrast, they had higher intakes of iron from fortified foods (8.8 [7.0–13.0] versus 5.7 [4.1–9.3] mg/day; $P = 0.005$), primarily from maize and wheat flour, which are staple foods in South Africa that are mandatorily fortified with iron.

## Study 3: efficacy and safety of iron supplementation with and without prebiotics

This randomized, double-blind, placebo-controlled trial assessed the efficacy and safety of 12 weeks of daily oral iron supplementation with versus without prebiotic GOS in iron-deficient children aged 10–15 years living with HIV. Participants received 50 mg of elemental iron as ferrous fumarate, consistent with the South African Standard Treatment Guidelines and Essential Medicines List[18]. The GOS dose (7.5 g) was selected based on prior evidence demonstrating its potential to enhance iron absorption from ferrous fumarate and mitigating adverse effects of iron fortification on the gut microbiome[19–21]. Participants were randomly assigned to receive iron+GOS ($n = 42$) or iron +placebo ($n = 44$). The primary outcome for sample size calculation was serum ferritin concentration, and co-primary outcomes were additional iron status biomarkers, including sTfR and Hb. Secondary outcomes were biomarkers of systemic and gut inflammation, gut integrity, HIV disease markers, targeted faecal bacteria, and adverse events and self-reported gastrointestinal and respiratory symptoms.

Recruitment began in September 2021 with study completion in October 2022 (Fig. 4). Of the 245 screened children with HIV, 86 were enrolled (iron+GOS: $n = 42$; iron+placebo: $n = 44$). Of these, 32 previously participated in Study 1 or 2. One child withdrew after the baseline visit, one after the 3-month visit, and two children were withdrawn from the study due to severe illness (serious adverse events; $n = 1$ in each treatment group). Eighty-three children

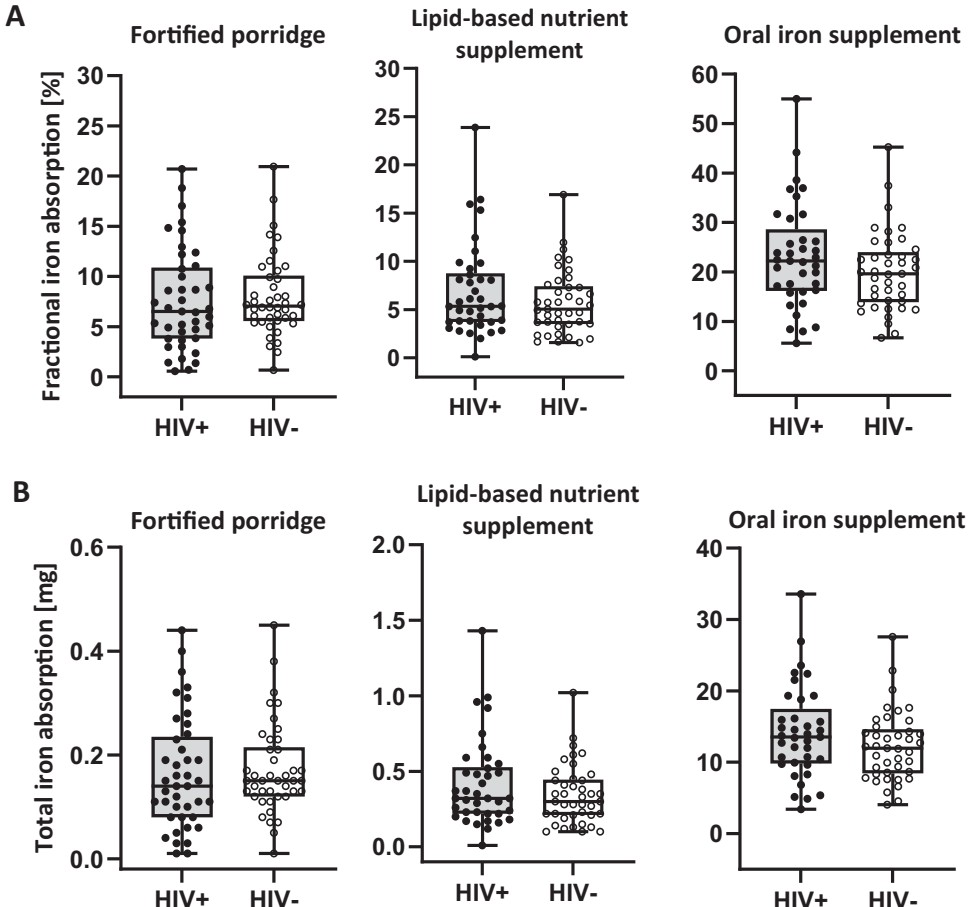

**Fig. 2 | Study 1: fractional iron absorption from three different iron interventions in children with virally suppressed HIV (HIV + , n = 43) and without HIV (HIV−, n = 45). A** Fractional iron absorption (%) and **B** total iron absorption (mg) from a maize porridge ( - 0.14 mg intrinsic iron per serving) extrinsically labelled with 2 mg $^{58}$Fe as ferrous fumarate (HIV − : n = 42; HIV + : n = 41), a lipid-based nutrient supplement (1.81 mg intrinsic iron per serving) extrinsically labelled with 6 mg $^{57}$Fe as ferrous sulphate (HIV − : n = 41; HIV + : n = 38), and a ferrous sulphate tablet containing 55 mg of elemental iron extrinsically labelled with 6 mg $^{57}$Fe as ferrous sulphate (HIV − : n = 41; HIV + : n = 37), in iron-depleted South African children with virally suppressed HIV and children without HIV. Boxes show the median and inter-quartile ranges; whiskers extend from the minimum to maximum value. Log-transformed FIA compared between groups using independent samples t-tests (two-sided). Median (IQR) FIA and total iron absorption did not differ between children with and without HIV for any of the three interventions (maize porridge, p = 0.256; LNS, p = 0.573; oral iron tablet, p = 0.264). No adjustments were made for multiple comparisons.

completed the trial (iron+GOS: n = 41; iron+placebo: n = 42). Median (IQR) compliance to the iron supplement was 95 (83–100)% in the iron +GOS group and 94 (88–98)% in the iron+placebo groups (P = 0.844). Compliance to the GOS and placebo powders was similarly high: 94 (87–100)% and 95 (85–100)% (P = 0.866), respectively.

The effect of the intervention on haemoglobin, biomakers of iron status, systemic and gut inflammation, gut integrity and markers of HIV disease are shown in Table 3. In both treatment groups, biomarkers of iron status, including haemoglobin, improved over the 12-month trial period (time, P < 0.001; all biomarkers). There was a trend toward a greater increase in serum ferritin in the iron+GOS group compared to the iron+placebo group (time x group, p = 0.053). Median (IQR) serum ferritin increased from 14.9 (9.0–23.4) to 40.8 (21.5–72.3) µg/L in the iron+GOS group, and from 13.6 (9.7–21.5) to 30.6 (16.5–50.2) µg/L in the iron+placebo group, corresponding to a 39% greater relative increase in the intervention group. However, no significant time x group interactions were observed for sTfR, haemoglobin, or the prevalence of either anaemia or iron deficiency.

There were also no significant time x group effects for plasma IFABP (p = 0.088) or faecal calprotectin (p = 0.533), a biomarker of gut inflammation. Changes in selected faecal bacteria over the 12-week period are shown in Table 4. No significant time x group effects were observed for any of the bacterial taxa or for a composite variable of the 6 pathogenic species. Self-reported respiratory and gastro-intestinal symptoms and side effects are shown in Table 5. The incidence of fever (incidence rate ratio [IRR] 0.14 [0.04-0.46], P = 0.001), rhinorrhoea (IRR 0.47 [0.25-0.88], P = 0.018), and flatulence (IRR 0.18 [0.07-0.46], P < 0.001) was lower in the iron+GOS compared to the iron+placebo group.

## Discussion

Our main findings are: (1) in Study 1, iron absorption from single doses of iron-fortified porridge, LNS and an oral iron supplement did not significantly differ between iron-depleted children with virally suppressed HIV and age- and sex-matched children without HIV; (2) in Study 2, over 6 months, children with virally suppressed HIV absorbed 24% less iron from their habitual diet than those without HIV; (3) in Study 3, iron-deficient children with virally suppressed HIV who received 50 mg oral iron daily with GOS for 12 weeks tended to have a 39% greater relative increase in serum ferritin (biomarker of iron stores) and a significantly lower incidence of fever and rhinorrhoea than those receiving iron with placebo.

To prevent and treat iron deficiency in children, experts and global health authorities recommend iron-fortified foods, iron-

**Table 2 | Study 2: Characteristics of the children with virally suppressed HIV and without HIV during the 6-month measurement period, showing values at baseline, midpoint (3 months) and endpoint (6 months) by group**

| | Children with HIV | Children without HIV | P value[a] | | |
|---|---|---|---|---|---|
| n | 29 | 36 | Time | Group | Time x group |
| Female/male sex assigned at birth, n (%) | 10/19 | 11/25 | - | 0.861 | - |
| **Age, y** | | | | | |
| Baseline | 12.0 (10.4–12.6) | 12.1 (10.8–13.3) | - | 0.394 | - |
| **Height-for-age Z-score** | | | | | |
| Baseline | −0.81 (−1.44–0.17) | −0.06 (−0.68–0.59) | - | 0.004 | - |
| **Weight-for-age Z-score** | | | | | |
| Baseline | −0.80 (−1.33– −0.14) | 0.19 (−0.70– 0.84) | - | <0.001 | - |
| **Body mass index for age Z score** | | | | | |
| Baseline | −0.63 (−0.86– −0.03) | 0.32 (−0.59– 0.88) | - | 0.003 | - |
| **Height, cm** | | | | | |
| Baseline | 138 (131–143) | 146 (135–153) | <0.001 | 0.016 | 0.907 |
| 3 months | 140 (133–145) | 147 (136–154) | | | |
| 6 months | 141 (135–146) | 149 (138–156) | | | |
| **Weight, kg** | | | | | |
| Baseline | 30.4 (27.5–34.2) | 37.4 (29.0–44.2) | <0.001 | 0.007 | 0.394 |
| 3 months | 31.5 (27.5–35.4) | 37.9 (29.7–46.9) | | | |
| 6 months | 33.0 (29.0–36.1) | 39.3 (30.0–48.3) | | | |
| **Haemoglobin, g/dL** | | | | | |
| Baseline | 12.4 (12.1–12.9) | 13.0 (12.3–13.6) | 0.013 | 0.173 | 0.621 |
| 3 months | 12.4 (12.1–12.6) | 12.6 (12.1–13.0) | | | |
| 6 months | 12.3 (11.9–12.9) | 12.3 (12.0–13.2) | | | |
| **Anaemia[b], n (%)** | | | | | |
| Baseline | 2 (7) | 4 (11) | 0.081 | 0.992 | 0.643 |
| 3 months | 3 (10) | 3 (8) | | | |
| 6 months | 6 (21) | 6 (17) | | | |
| **Serum ferritin, adjusted[c], µg/L** | | | | | |
| Baseline | 32.8 (23.6–42.6) | 29.6 (23.1–43.7) | 0.559 | 0.579 | 0.005 |
| 3 months | 38.2 (29.4–44.5) | 30.0 (21.2–38.1) | | | |
| 6 months | 31.7 (23.2–45.9) | 35.5 (26.3–47.8) | | | |
| **Soluble transferrin receptor, mg/L** | | | | | |
| Baseline | 6.64 (5.93–8.20) | 6.36 (5.52–7.73) | 0.502 | 0.058 | 0.168 |
| 3 months | 6.72 (5.79–8.20) | 6.49 (5.57–7.59) | | | |
| 6 months | 7.16 (6.27–8.00) | 6.32 (5.50–7.21) | | | |
| **Iron deficiency[d], n (%)** | | | | | |
| Baseline | 6 (21) | 4 (11) | 0.477 | 0.161 | 0.497 |
| 3 months | 7 (24) | 6 (17) | | | |
| 6 months | 8 (28) | 4 (11) | | | |
| **C-reactive protein, mg/L** | | | | | |
| Baseline | 0.20 (0.02–0.84) | 0.04 (0.03–1.53) | <0.001 | 0.634 | 0.067 |
| 3 months | 0.15 (0.03–0.86) | 0.04 (0.02–0.11) | | | |
| 6 months | 0.23 (0.05–0.97) | 0.15 (0.04–0.33) | | | |
| **Alpha-1-glycoprotein, g/L** | | | | | |
| Baseline | 0.54 (0.45–0.63) | 0.53 (0.43–0.76) | <0.001 | 0.195 | 0.027 |
| 3 months | 0.57 (0.44–0.69) | 0.44 (0.37–0.53) | | | |
| 6 months | 0.60 (0.45–0.79) | 0.50 (0.38–0.60) | | | |
| **Inflammation[e], n(%)** | | | | | |
| Baseline | 3 (10) | 6 (17) | <0.001 | <0.001 | - |
| 3 months | 3 (11) | 0 | | | |
| 6 months | 5 (17) | 0 | | | |
| **Hepcidin** | | | | | |
| Baseline | 4.45 (2.63–7.79) | 6.59 (3.08–9.68) | 0.393 | 0.888 | 0.220 |
| 3 months | 5.30 (2.28–10.51) | 4.18 (2.55–6.57) | | | |
| 6 months | 5.66 (2.03–10.13) | 5.24 (3.09–8.19) | | | |

Values are medians (IQR) or n (%).

[a]Time, group and time x group interactions were analysed using repeated-measures ANOVA for continuous outcome variables (two-sided; non-normally distributed variables were log-transformed for analysis) and generalized estimating equations (two-sided) for categorical outcome variables. No adjustments were made for multiple comparisons.

[b]Defined as haemoglobin <11.5 g/dL for children aged <12 years, and <12.0 g/dL for those aged ≥12 years.

[c]Serum ferritin adjusted according to the BRINDA recommendations[30].

[d]Defined as adjusted serum ferritin <15 µg/L and/or soluble transferrin receptor >8.5 mg/L.

[e]Defined as C-reactive protein >5 mg/L and/or alpha-1-glycoprotein >1 g/L.

containing LNS and oral iron supplements[4,16,17,22]. However, because chronic low-grade inflammation, characteristic of ART-treated HIV[23] may limit iron absorption[24], it is uncertain whether these strategies should be recommended for children with HIV. Our findings suggest that these strategies may still be effective, particularly in virally suppressed children receiving ART.

In study 1, children with HIV had higher median sTfR concentrations (indicator of iron-deficient erythropoiesis) and more anaemia than children without HIV, despite both groups being enrolled based on low ferritin concentrations. Although the children with HIV had more systemic inflammation, they had 45% lower hepcidin concentrations than children without HIV. The hormone hepcidin,

produced by the liver, reduces iron absorption by binding and degrading the iron-exporter ferroportin on enterocytes[25]. It appears that hepcidin suppression by iron deficiency and erythropoiesis overrode the inflammatory stimulus to increase hepcidin synthesis[25] and this resulted in comparable iron absorption between the two groups. A similar pattern of hierarchal control of hepcidin has been described in iron-deficient adult women without HIV[26]. However, in other contexts, such as acute infections like malaria and tuberculosis, co-existing inflammation may dominate and elevate hepcidin despite underlying iron deficiency[27,28].

In study 2, we applied a novel isotope method previously used in Gambian and Malawian children without HIV[13,29]. The method allows for direct measurement of dietary iron absorption and loss over extended periods and may be particularly useful during HIV infection, because chronic inflammation may skew common iron status biomarkers[12]. In this method, dietary iron of natural composition acts as a tracer diluting an ad hoc-modified isotopic signature obtained via isotope administration and equilibration with body iron. Over the 6-month measurement period, the median iron absorption in children with HIV (1.09 mg/day) was below the median absorbed iron requirement for children of this age, which is 1.17-1.20 mg/day[30]. In contrast, the median iron absorption in children without HIV (1.44 mg/day) was above the requirement. This likely explains the higher median sTfR concentrations and the lack of increased ferritin concentrations over the 6-month period in children with HIV compared to those without HIV. The longitudinal prevalence of inflammation was higher in the children with HIV. Furthermore, their diet was lower in bioavailable haem iron and vitamin C which enhances non-haem iron-absorption (Supplementary Table 3). Therefore, the lower iron absorption observed in children with HIV is likely attributed to a combination of inflammatory and dietary factors, specifically low dietary iron bioavailability, which may be underpinned by socio-economic disparities between children with and without HIV (Supplementary Table 4). ART-treated HIV is further characterized by persistent viral replication in the gut mucosa[6–8], and children with HIV in study 1 had borderline higher plasma IFAPB (a measure of enterocyte damage)[31] (Table 1). However, our data suggest persistent mucosal HIV infection does not significantly increase gastrointestinal iron loss, as median iron losses were not statistically greater in the children with HIV, and there were no significant differences in net iron balance between the groups.

Although iron deficiency is common in children with HIV[2], clinicians may be reluctant to give iron to children with HIV due to

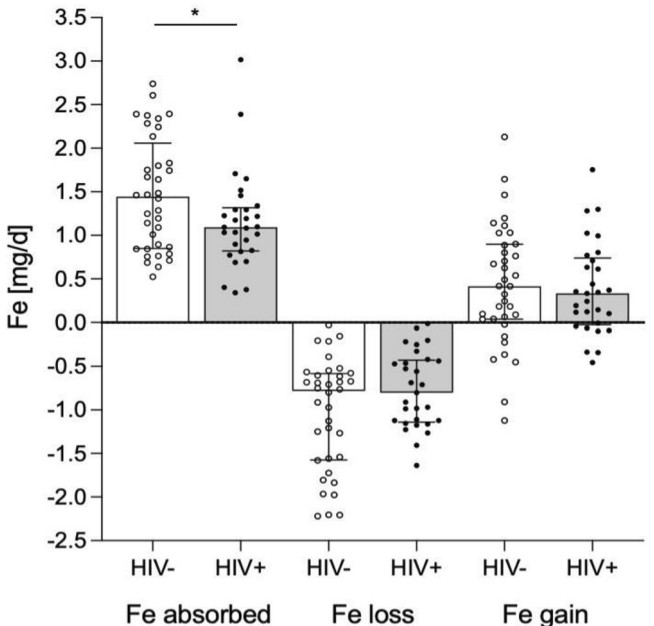

**Fig. 3 | Study 2: Amount of absorbed iron, iron loss and net iron gains (mg/day) in South African children living with virally suppressed HIV (HIV + , *n* = 29) and children living without HIV (HIV-, *n* = 36) over 6 months.** Data shown as median (IQR). Compared using independent samples t-test (two-sided) on log-transformed data. Median dietary iron absorption (mg/day) was significantly lower in children with compared to without HIV (*$p$ = 0.042).

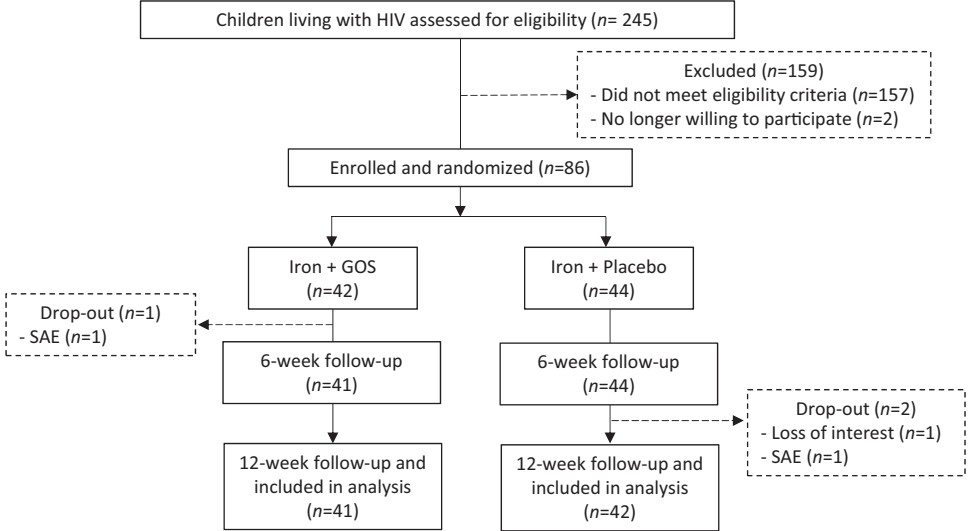

**Fig. 4 | Design and participant flow diagram for Study 3.** SAE serious adverse event.

**Table 3 | Study 3: Effect of prebiotic galacto-oligosaccharides as adjunct treatment to 12 weeks of oral iron supplementation on iron status, systemic and gut inflammation, gut integrity and markers of HIV disease in iron-deficient children with virally suppressed HIV**

| | week | n | Iron + GOS | n | Iron + Control | P-value[a] Time | Group | Time x group |
|---|---|---|---|---|---|---|---|---|
| Age, y | 0 | 41 | 14 (12, 15) | 42 | 14 (12, 14) | - | 0.169 | - |
| Female/male sex assigned at birth, n | 0 | 41 | 27/14 | 42 | 19/23 | - | 0.001 | - |
| Height, cm | 0 | 41 | 149 (143, 155) | 42 | 150 (143, 158) | - | 0.808 | - |
| Weight, kg | 0 | 41 | 40.6 (33.6, 47.5) | 42 | 41.6 (33.0, 50.5) | - | 0.999 | - |
| Haemoglobin, g/dL | 0 | 41 | 11.9 (11.2, 12.7) | 42 | 12.2 (11.4, 13.0) | <0.001 | 0.197 | 0.961 |
| | 6 | 41 | 12.3 (11.5, 13.1) | 42 | 12.4 (12.2, 13.0) | | | |
| | 12 | 36 | 12.6 (11.7, 13.4) | 38 | 12.5 (12.1, 14.1) | | | |
| Mean corpuscular volume, fL | 0 | 41 | 93.4 (85.6, 96.9) | 42 | 92.0 (87.7, 95.1) | 0.850 | 0.656 | 0.989 |
| | 6 | 41 | 92.5 (86.5, 97.1) | 42 | 90.1 (88.2, 94.4) | | | |
| | 12 | 36 | 91.7 (85.5, 98.8) | 38 | 91.1 (87.6, 97.0) | | | |
| Ferritin, µg/L | 0 | 40 | 14.9 (9.0, 23.4) | 40 | 13.6 (9.7, 21.5) | <0.001 | 0.002 | 0.053 |
| | 6 | 41 | 30.8 (20.1, 49.2) | 41 | 28.6 (17.6, 38.3) | | | |
| | 12 | 40 | 40.8 (21.5, 72.3) | 40 | 30.6 (16.5, 50.2) | | | |
| Soluble transferrin receptor, mg/L[3] | 0 | 40 | 7.4 (5.4, 9.8) | 41 | 7.3 (6.1, 8.4) | <0.001 | 0.769 | 0.532 |
| | 6 | 41 | 6.2 (5.2, 7.7) | 41 | 6.4 (5.8, 7.2) | | | |
| | 12 | 40 | 5.3 (4.9, 7.3) | 40 | 6.2 (5.2, 6.7) | | | |
| Iron deficiency, n (%) | 0 | 40 | 26 (65) | 40 | 27 (68) | <0.001 | 0.037 | 0.752 |
| | 6 | 41 | 10 (24) | 41 | 13 (32) | | | |
| | 12 | 40 | 12 (30) | 40 | 14 (35) | | | |
| Body iron stores, mg/kg | 0 | 40 | 1.3 (–1.2, 3.2) | 41 | 0.6 (–0.8, 2.8) | <0.001 | 0.005 | 0.599 |
| | 6 | 41 | 4.7 (2.6, 6.3) | 41 | 3.9 (1.6, 5.4) | | | |
| | 12 | 40 | 5.7 (3.2, 8.1) | 40 | 4.7 (2.0, 6.1) | | | |
| C-reactive protein, mg/L | 0 | 40 | 0.24 (0.06, 1.08) | 41 | 0.20 (0.08, 0.96) | 0.165 | 0.581 | 0.921 |
| | 6 | 41 | 0.30 (0.16, 1.20) | 41 | 0.19 (0.09, 1.19) | | | |
| | 12 | 40 | 0.40 (0.15, 1.08) | 40 | 0.42 (0.15, 1.12) | | | |
| α-1-acid glycoprotein, g/L | 0 | 40 | 0.68 (0.58, 0.79) | 41 | 0.63 (0.53, 0.78) | 0.097 | 0.198 | 0.900 |
| | 6 | 41 | 0.66 (0.54, 0.77) | 41 | 0.62 (0.51, 0.82) | | | |
| | 12 | 40 | 0.70 (0.61, 0.83) | 40 | 0.71 (0.58, 0.81) | | | |
| Subclinical inflammation, n (%) | 0 | 40 | 0 (0) | 41 | 0 (0) | 0.136 | 0.248 | 0.136 |
| | 6 | 41 | 0 (0) | 41 | 0 (0) | | | |
| | 12 | 40 | 0 (0) | 40 | 2 (5) | | | |
| Inflammation, n (%) | 0 | 40 | 0 (0) | 41 | 2 (5) | 0.209 | 0.488 | 0.531 |
| | 6 | 41 | 2 (5) | 41 | 4 (10) | | | |
| | 12 | 40 | 4 (10) | 40 | 0 (0) | | | |
| Hepcidin, µg/L | 0 | 40 | 1.45 (0.79, 2.87) | 40 | 1.43 (0.81, 3.49) | <0.001 | 0.262 | 0.706 |
| | 6 | 38 | 3.69 (1.76, 5.95) | 41 | 3.19 (1.71, 5.81) | | | |
| | 12 | 40 | 5.43 (1.97, 9.71) | 42 | 4.33 (1.78, 7.80) | | | |
| I-FABP, pg/mL | 0 | 39 | 602 (437, 870) | 42 | 663 (418, 883) | 0.620 | 0.995 | 0.088 |
| | 12 | 39 | 583 (430, 952) | 41 | 788 (519, 1090) | | | |
| Faecal calprotectin, mg/kg | 0 | 41 | 19.2 (8.4, 58.3) | 42 | 19.0 (7.1, 58.3) | 0.541 | 0.538 | 0.533 |
| | 12 | 40 | 32.7 (11.7, 57.9) | 42 | 21.0 (7.5, 47.2) | | | |
| Faecal pH | 0 | 41 | 7.0 (6.4, 7.1) | 42 | 6.8 (6.3, 7.1) | 0.734 | 0.743 | 0.845 |
| | 12 | 41 | 6.9 (6.3, 7.2) | 42 | 6.8 (6.2, 7.1) | | | |
| CD4 + T cell count, counts/mm | 0 | 24 | 713.5 (553.3, 1145.5) | 29 | 669.0 (568.5, 774.5) | 0.069 | 0.469 | 0.259 |
| | 12 | 34 | 625.5 (535.0, 806.3) | 33 | 671.0 (514.5, 798.5) | | | |
| CD4 + T cells, % | 0 | 24 | 37.5 (31.9, 41.4) | 29 | 37.0 (29.7, 40.3) | 0.958 | 0.912 | 0.858 |
| | 12 | 34 | 36.2 (32.0, 40.9) | 33 | 34.2 (30.6, 38.9) | | | |
| CD4 + T cell count <500, n | 0 | 24 | 4 | 29 | 4 | 0.565 | 0.768 | 0.663 |
| | 12 | 34 | 6 | 33 | 6 | | | |
| Detectable viral load >20 copies/mL, n | 0 | 41 | 9 | 42 | 8 | 0.101 | 0.121 | 0.399 |
| | 12 | 40 | 7 | 41 | 2 | | | |

Values are medians (IQR) or n (%).
[a] Time, group and time x group interactions were analysed using general linear models (two-sided), adjusted for sex and age. For ferritin, AGP and CRP were added as additional covariates. No adjustments were made for multiple comparisons. Exact p-values are reported where possible.

**Table 4 | Study 3: Selected bacteria in faecal samples of iron-deficient South African children with virally suppressed HIV before and after supplementation with iron or iron and galacto-oligosaccharides (GOS) daily for 12 weeks**

| | week | Iron+GOS | | Iron+Control | | P-value[a] | | |
|---|---|---|---|---|---|---|---|---|
| | | n | log gene copies/total 16S rRNA gene copies | n | log gene copies/total 16S rRNA gene copies | Time | Group | Time x group |
| *Bifidobacterium* spp. | 0 | 41 | −1.1169 (−1.4446, −0.8438) | 42 | −1.2310 (−2.0760, −0.8129) | 0.627 | <0.001 | 0.755 |
| | 12 | 41 | −0.9594 (−1.4648, −0.6734) | 42 | −1.2178 (−2.6869, −0.8666) | | | |
| *Enteropathogenic Escherichia coli* (EPEC) | 0 | 41 | −3.0342 (−3.2164, −2.6422) | 42 | −2.9318 (−3.2596, −2.3609) | 0.909 | 0.958 | 0.218 |
| | 12 | 41 | −2.8318 (−3.2914, −2.1491) | 42 | −3.0527 (−3.2781, −2.5577) | | | |
| *Enterotoxigenic Escherichia coli* (ETEC) | 0 | 41 | −4.2928 (−4.5662, −3.9031) | 42 | −4.2587 (−4.5621, −3.7426) | 0.400 | 0.906 | 0.484 |
| | 12 | 41 | −4.1531 (−4.5459, −3.8013) | 42 | −4.2858 (−4.5954, −3.9915) | | | |
| *Campylobacter* spp. | 0 | 41 | −4.7386 (−5.8079, −4.4008) | 42 | −4.7339 (−5.3057, −4.1267) | 0.248 | 0.109 | 0.413 |
| | 12 | 41 | −5.0535 (−5.9813, −4.5091) | 42 | −4.7288 (−5.8861, −4.3948) | | | |
| *Salmonella* spp. | 0 | 41 | −4.8008 (−4.9707, −4.4753) | 42 | −4.7086 (−5.0616, −4.1253) | 0.666 | 0.765 | 0.390 |
| | 12 | 41 | −4.5783 (−5.0339, −4.2619) | 42 | −4.7149 (−4.9890, −4.3656) | | | |
| *Clostridium difficile* | 0 | 41 | −4.2170 (−4.4217, −3.9088) | 42 | −4.1576 (−4.4682, −3.5997) | 0.905 | 0.864 | 0.306 |
| | 12 | 41 | −3.9078 (−4.3993, −3.6681) | 42 | −4.1114 (−4.4089, −3.8071) | | | |
| *Clostridium perfringens* | 0 | 41 | −2.7846 (−3.1401, −2.3314) | 42 | −2.8589 (−3.2478, −2.4016) | 0.248 | 0.480 | 0.607 |
| | 12 | 41 | −2.6163 (−3.0298, −2.0907) | 42 | −2.6649 (−3.1946, −2.1209) | | | |
| Summed variable of six pathogens | 0 | 41 | −2.4569 (−2.6401, −2.0565) | 42 | −2.4713 (−2.8504, −2.0424) | 0.551 | 0.513 | 0.897 |
| | 12 | 41 | −2.2967 (−2.7273, −1.7221) | 42 | −2.3197 (−2.8746, −1.9250) | | | |

Values are medians (IQR).

[a]Time, group, and time x group interactions were analysed using linear-mixed models (two-sided) using the log transformed ratio of each target as dependent variable, group and time as fixed effects, study IDs as random effects, and sex, age, faecal pH, and faecal calprotectin as covariates. No adjustments were made for multiple comparisons.

concerns about increased infection risk[31,32]. In a trial of iron supplementation (3 mg/kg/day for 3 months) in 6–59-month-old anaemic Malawian children with HIV ($n = 209$), iron treatment increased haemoglobin and reduced the risk of anaemia but increased the incidence of malaria[33]. However, only a third of the children were on ART. In a recent 3-month trial of iron supplements (12.5-30.0 mg/day) in Ugandan children aged 6 months to 12 years with HIV ($n = 204$; 40% anaemic; all on ART) who were provided anti-malarial prophylaxis and bed nets, iron treatment increased ferritin without increasing morbidity[9]. A non-randomised 6-month study of iron supplementation of Indian children living with HIV, 44% of whom were on ART, reported improvements in haemoglobin and no increased risk of infection[10]. In contrast, a cohort study in Tanzanian children with HIV reported an increase in infections with iron supplementation[34]. Providing lower iron doses and maximizing iron absorption may be important in children with HIV because high amounts of residual, unabsorbed iron in the intestine causes mucosal inflammation[35], damages intestinal enterocytes and causes adverse shifts in the gut microbiome, increasing gram-negative enteropathogens[36] and diarrhea[37]. In a subsample of 70 children in the Ugandan trial[9], there were no major effects of iron treatment on the gut microbiome assessed by 16S rRNA sequencing, but the authors suggested they were underpowered for this outcome. In healthy African infants, prebiotic fibres, such as GOS, can modulate gut immunity and increase iron absorption[14,20]. In our study, iron-deficient children with HIV who received 50 mg oral iron daily with GOS for 12 weeks showed a borderline-significant 39% greater relative increase in serum ferritin compared to those who received oral iron with placebo. The GOS group also had a lower incidence of fever and rhinorrhoea, while no significantly differences were observed in biomarkers of gut inflammation, integrity or pathogen abundances. If confirmed by larger studies and validated in children not receiving ART or living in malaria-endemic regions, these findings suggest prebiotics may improve both the efficacy and safety of oral iron supplementation in iron-deficient children with HIV.

A key strength of our study is the application of both single-dose and long-term quantitative isotopic techniques to investigate iron absorption and loss in virally suppressed children with HIV, alongside age- and sex-matched HIV-negative controls (Studies 1 and 2). The three studies differed in design and duration. Study 1 provides insights into short-term iron absorption from commonly-used iron interventions under controlled conditions, whereas Study 2 captures longer-term iron absorption from the habitual diet—including fortified foods—as well as endogenous losses under real-world dietary and inflammatory conditions. Together, these complementary approaches offer a more comprehensive and physiologically relevant understanding of iron kinetics in this population.

However, several limitations should be acknowledged. First, our findings are likely generalizable only to children with HIV receiving effective and well-monitored ART, which unfortunately remains inaccessible to many in rural areas of sub-Saharan Africa[38]. Second, in Study 1, iron absorption was assessed following a single administration of each intervention under fasting conditions. Such single-dose studies may not fully represent habitual absorption patterns and may overestimate or underestimate an individual's true iron absorption capacity. Third, the inclusion of younger children in Study 2 would have been preferable to minimise potential confounding from the onset of menarche and the increased physiological iron demands of adolescent growth. However, we addressed this by excluding data from girls who had begun menstruating. While this reduced potential bias, it also led to a smaller sample size and decreased statistical power. Finally, Study 3 lacked an iron-placebo control arm and was conducted in a malaria-free setting, which may limit its generalizability to regions where malaria is endemic.

In conclusion, our findings demonstrate that iron-depleted children with virally suppressed HIV absorb iron from fortified foods and supplements as effectively as children without HIV, and do not have increased gastrointestinal iron losses. These results are reassuring, suggesting that iron interventions are safe and effective in children

**Table 5 | Study 3: Effect of prebiotic galacto-oligosaccharides (GOS) as adjunct treatment to 12 weeks of oral iron supplementation on self-reported respiratory and gastro-intestinal symptoms and side effects**

| | Iron + GOS<br>N participants = 31 | Iron + Placebo<br>N participants = 29 | OR (95% CI)[a] / IRR (95% CI)[b] | P-value |
|---|---|---|---|---|
| **Fever** | | | | |
| Participants with symptom, *n (%) children* | 3 (10) | 9 (31) | 0.24 (0.06, 1.05) | 0.058 |
| Episodes with symptom, *n (%) events* | 3 (0.13) | 20 (0.93) | 0.14 (0.04, 0.46) | 0.001 |
| **Coughing** | | | | |
| Participants with symptom, *n (%) children* | 8 (26) | 8 (28) | 1.06 (0.32, 3.45) | 0.926 |
| Episodes with symptom, *n (%) events* | 21 (0.92) | 28 (1.30) | 0.81 (0.46, 1.44) | 0.474 |
| **Rhinorrhoea** | | | | |
| Participants with symptom, *n (%) children* | 8 (26) | 7 (24) | 1.21 (0.36, 4.04) | 0.760 |
| Episodes with symptom, *n (%) events* | 14 (0.61) | 33 (1.53) | 0.47 (0.25, 0.88) | 0.018 |
| **Diarrhoea** | | | | |
| Participants with symptom, *n (%) children* | 6 (19) | 8 (28) | 0.66 (0.19, 2.35) | 0.522 |
| Episodes with symptom, *n (%) events* | 8 (0.35) | 12 (0.56) | 0.61 (0.24, 1.55) | 0.301 |
| **Stomach ache/abdominal pain** | | | | |
| Participants with symptom, *n (%) children* | 9 (29) | 9 (31) | 0.87 (0.28, 2.72) | 0.805 |
| Episodes with symptom, *n (%) events* | 27 (1.18) | 27 (1.25) | 1.02 (0.59, 1.77) | 0.934 |
| **Nausea** | | | | |
| Participants with symptom, *n (%) children* | 7 (23) | 6 (21) | 1.11 (0.32, 4.03) | 0.843 |
| Episodes with symptom, *n (%) events* | 24 (1.05) | 15 (0.70) | 1.18 (0.60, 2.32) | 0.632 |
| **Vomiting** | | | | |
| Participants with symptom, *n (%) children* | 4 (13) | 2 (7) | 1.74 (0.28, 10.9) | 0.555 |
| Episodes with symptom, *n (%) events* | 7 (0.30) | 3 (0.14) | 1.55 (0.38, 6.38) | 0.542 |
| **Constipation** | | | | |
| Participants with symptom, *n (%) children* | 4 (13) | 4 (14) | 0.77 (0.16, 3.72) | 0.771 |
| Episodes with symptom, *n (%) events* | 5 (0.22) | 9 (0.42) | 0.52 (0.17, 1.59) | 0.250 |
| **Flatulence** | | | | |
| Participants with symptom, *n (%) children* | 5 (16) | 10 (35) | 0.37 (0.11, 1.13) | 0.126 |
| Episodes with symptom, *n (%) events* | 5 (0.22) | 29 (1.34) | 0.18 (0.07, 0.46) | <0.001 |
| **Headache** | | | | |
| Participants with symptom, *n (%) children* | 6 (19) | 4 (14) | 1.32 (0.29, 5.95) | 0.715 |
| Episodes with symptom, *n (%) events* | 10 (0.44) | 4 (0.19) | 2.36 (0.69, 8.11) | 0.173 |

[a]The proportion (%) of participants who had at least one episode of the respective symptom during the intervention period was compared between the two treatment groups using logistic regression analysis (two-sided), adjusted for age and sex. Results are expressed as Odds Ratio [OR] with 95% confidence interval [CI]).

[b]The incidence (number of episodes) with symptoms was compared between the two treatment groups using Poisson regression with log-linear link function (two-sided), adjusted for age and sex (expressed as Incidence Rate Ratio [IRR] with 95% CI). No adjustments were made for multiple comparisons.

living with virally suppressed HIV. Moreover, our findings highlight the potential of prebiotics to further enhance treatment outcomes. Collectively, this work provides a foundation for evidence-based recommendations to prevent and treat iron deficiency and anaemia in this vulnerable population.

## Methods
### Study 1: design and participants
Study 1 followed a prospective crossover design with case-control comparisons in iron-depleted children with HIV ($n = 45$) and without HIV ($n = 45$). Children with HIV were recruited from the Tygerberg Hospital infectious disease outpatient unit in Cape Town or previous HIV research cohorts, and children without HIV were recruited from similar communities by community outreach teams. We screened primary school-aged children (8–13 years) who met the following inclusion criteria: (1) for children with virally suppressed HIV: (i) receiving ART; (ii) HIV plasma viral load <50 copies/mL (from laboratory records); (iii) serum ferritin ≤40 μg/L; and (2) for children without HIV: (i) HIV-negative based on rapid HIV assay (First Response HIV Card 1-2.0, Premier Medical Corporation Pvt Ltd, Sarigam, India); (ii) serum

ferritin ≤40 μg/L. Exclusion criteria for both groups were: (i) acute illness, (ii) receiving iron supplements in the previous three months, (iii) severe over- or undernutrition (a BMI-for-age Z-score <−3 or >2)[39], (iv) severe anaemia (haemoglobin ≤80 g/L)[40], or (v) self-reported peanut or milk allergy (ingredients in the LNS). The two groups were matched for sex and age. For Study 1, we included children with depleted iron stores by applying a serum ferritin threshold of ≤40 μg/L, based on recent evidence indicating that iron absorption begins to increase at ferritin concentrations around 45 μg/L in both infants and women of reproductive age[41,42]—a physiological response to early iron deficiency. Also, most recent WHO guidelines recommend a ferritin cut-off of <70 μg/L for defining iron deficiency in children ≥5 years with inflammation[43], further supporting the use of a higher threshold.

Children with and without HIV who met the inclusion criteria but had a serum ferritin >40 μg/L ($n = 90$) were later invited to participate in Study 2.

The study design is shown in Fig. 1. At screening, we measured body weight and height, and collected socio-economic and demographic information, and venous blood for measurement of haemoglobin, serum ferritin, soluble transferrin receptor [sTfR], hepcidin,

systemic inflammation (C-reactive protein [CRP], α−1-acid glycoprotein [AGP]) and viral loads in children with HIV. After an overnight fast, all enrolled children were administered the three iron interventions on three non-consecutive days under supervision. On days 1 and 3, the children consumed either a maize porridge labelled with 2 mg of $^{58}$Fe as ferrous fumarate or an LNS labelled with 6 mg of $^{57}$Fe as ferrous sulphate, both given with 250 mL bottled water. Their order was randomized for each child using a computer-generated list (Excel, Microsoft Office 2016).

On day 17, we collected venous blood to determine FIA from the first two iron interventions, and a ferrous sulphate tablet containing 55 mg of elemental iron (Gulf Drug Company, South Africa) labelled with 6 mg of $^{57}$Fe as ferrous sulphate was given with 250 mL of bottled water. After each administration, the children remained fasted at the study site for a further two hours before consuming a standardized lunch. We again collected a venous blood sample at day 31, the final study visit, to determine FIA from the third iron intervention. Children with a serum ferritin ≤35 μg/L (local clinical threshold) at endpoint received 3 months of oral iron supplementation as per standard of care.

## Study 1: maize porridge and lipid-based nutrient supplement composition

The maize porridge was cooked fresh on the day of administration. A portion was 140.0 g and consisted of 20.0 g unfortified maize flour (TAU Mills, Leeudoringstad, South Africa), 28.0 mL full cream milk (Parmalat EverFresh UHT, South Africa), 108.5 mL natural spring water (Aquellé, South Africa), 7.0 g sugar (Illovo Sugar, South Africa), and 0.4 g salt (Cerebos, South Africa). The LNS portion was 55.0 g; it was produced by the study team and consisted of 13.2 g canola oil (Sunfoil, South Africa), 12.7 g peanut paste (Black Cat, Tiger Brands, South Africa), 12.7 g full cream instant milk powder (Klim, Nestlè, Switzerland), 3.8 g sugar (Illovo Sugar Pty Ltd, South Africa), 8.3 g maltodextrin (Nutritional Performance Labs, South Africa), and 4.4 g palm stearin (Florin AG, Switzerland). In maize flour and the LNS, we measured the native iron concentration using an inductively coupled plasma atomic emission spectrophotometer (ICP-AES; Varian VistaPro), measured phytic acid concentrations using a modification of the method of Makower[44], in which the released inorganic phosphate is measured colorimetrically[45] and converted into phytic acid concentrations. We used a modified Folin-Ciocalteau method[46] to measure total polyphenol concentration of the maize flour and LNS. The native iron concentration in the maize flour was 0.50 ± 0.02 mg/100 g dry matter (~ 0.14 mg/test meal), and in the LNS was 3.30 ± 0.04 mg/100 g dry matter (~ 1.81 mg/test meal). Together with the 2 mg $^{58}$Fe as ferrous fumarate, the total iron content in one maize porridge portion amounted to 2.14 mg, and together with the 6 mg $^{57}$Fe as ferrous sulphate, the total iron content in one LNS portion amounted to 7.81 mg. The phytic acid and polyphenol concentrations in the maize flour were 175.2 ± 15.1 mg/100 g dry matter (~ 49.0 mg/test meal) and 91.3 ± 5.30 mg/100 g dry matter (~ 25.6 mg/test meal), respectively. The phytic acid and polyphenol concentrations in the LNS were 9.8 ± 4.12 mg/100 g dry matter (~ 130.10 mg/test meal) and 107.4 ± 10.4 mg/100 g dry matter (~ 59.1 mg/test meal), respectively. The phytic acid: iron molar ratio in the maize porridge and the LNS were 1:1, and 0.7:1, respectively.

## Study 2: design and participants

We enrolled the participants of Study 1 and an additional 90 children (8–13 years) with ($n = 45$) and without ($n = 45$) HIV in Study 2, a 14-month long-term isotope dilution study (Fig. 1). The inclusion and exclusion criteria for the additional 90 participants were the same as in Study 1, except that we also included children with serum ferritin ≥40 μg/L. A single trained dietitian interviewed the child-caregiver pairs to assess intake of selected nutrients/dietary components over

the previous month using an abbreviated quantified food frequency questionnaire validated for this study population[47] (detail in Supplementary Methods). All children ($n = 90$ with HIV; $n = 90$ without HIV) were labelled with a total of 12 mg $^{57}$Fe, either by taking part in Study 1 or by consuming a single dose of 12 mg $^{57}$Fe as ferrous sulphate after an overnight fast. All participants then entered an 8-month isotopic equilibration period during which the $^{57}$Fe distributed uniformly throughout body iron[12]. At the end of the 8-month equilibration period, we followed the children for additional 6 months (measurement period) to measure isotopic dilution for quantification of daily dietary iron absorption, and iron losses and gains. In total, we collected five fasting venous blood samples to determine isotopic composition: at 0, 4 and 8 months of the equilibration period (the final sample was used as the baseline of the measurement period) and at 3 and 6 months of the measurement period (Fig. 1). We also measured haemoglobin, serum ferritin, sTfR, CRP, AGP and hepcidin in the three blood samples collected in the measurement period.

## Study 3: design and participants

Study 3 was a randomized, double-blind, placebo-controlled trial of GOS supplementation as adjunct treatment to 12 weeks of oral iron (50 mg as ferrous fumarate) in iron-deficient children with HIV (Fig. 2) with serum ferritin and additional biomarkers of iron status, including sTfR and haemoglobin as the primary outcomes. Secondary outcomes of safety were: (i) biomarkers of systemic and gut inflammation, and gut mucosal integrity; (ii) viral load and CD4 T cell counts; (iii) targeted faecal bacteria; and (iv) adverse events and self-reported gastrointestinal and respiratory symptoms. Participants were recruited from the participants with HIV in Study 1 and/or 2 ($n = 32$). Accordingly, the age range for inclusion in this study was set at 10–15 years to match the ages of these participants at the time of recruitment. Additional participants ($n = 56$) with HIV were recruited by FAMCRU at Tygerberg Hospital in Cape Town. Inclusion criteria were: (i) 10–15 years of age; (ii) living with HIV and receiving ART with plasma HIV viral load <50 copies/mL; (iv) iron-deficient, defined as serum ferritin <30 μg/L or sTfR >8.3 mg/L, and/or normocytic or microcytic anaemic, defined as haemoglobin <110/120 (children 10–11/12–15 years) g/L and mean corpuscular volume (MCV) < 91.5 fL (cut-off value used by the National Health Laboratory Service of South Africa). Exclusion criteria were: (i) acute illness, (ii) use of iron supplements or antibiotics in the previous three months, (iii) severe malnutrition (defined as in Study 1), and (iv) severe anaemia (haemoglobin ≤80 g/L).

Participants were randomly allocated to receive: 1) 50 mg oral iron as ferrous fumarate (Rulofer N, Lomapharm GmbH, Hannover, Germany) with 7.5 g of GOS ($n = 44$) or 2) 50 mg oral iron as ferrous fumarate with placebo (maltodextrin) ($n = 44$) daily for 12 weeks at a ratio of 1:1 and stratified by sex and age using the RANNOR function of the SAS software package (version 9.4). The 7.5 g GOS was provided in sachets containing 10.5 g powder (75% GOS; Vivinal GOS 75 Powder; FrieslandCampina, Amersfoort, Netherlands); this GOS dose increased iron absorption in children in a previous study[20]. The children in the placebo group received sachets containing 10.5 g of maltodextrin powder. We instructed the caregivers to dissolve a sachet in ~200 ml of water before consumption together with the iron supplement. The two powders had similar appearance and taste. Researchers, research staff and subjects were blinded to the intervention, codes were held by an independent researcher not involved in the study, and the codes were broken only after the data analyses were completed. At baseline, 6 and 12 weeks, we collected venous blood, and at baseline and 12 weeks, we collected faecal samples, which were aliquoted and frozen. Supplement compliance and gastrointestinal and respiratory symptoms were recorded daily by the caregiver using a diary, and unused sachets were collected by the field team before each new disbursement. Adverse events were recorded by fieldworkers during monthly calls. In blood, we measured (i) haemoglobin, MCV, serum ferritin, sTfR, plasma iron,

total iron binding capacity (TIBC), transferrin saturation (Tsat), body iron stores, hepcidin, (ii) systemic inflammation (CRP and AGP), (iii) gut mucosal integrity (plasma IFABP), and (iv) viral loads and CD4 T cell counts. In the sub-group of isotopically-equilibrated children, we determined iron absorbed and lost during supplementation by measuring isotopic dilution. In faecal samples, we measured pH and calprotectin, a biomarker of gut inflammation, and using qPCR, we measured *Bifidobacterium spp*. and the virulence and toxin genes of six enteropathogenic bacteria (Supplemental Material).

## Anthropometric measurements and laboratory analyses

We measured body weight to the nearest 0.1 kg and height to the last completed 0.1 cm, using a Micro 1023 electronic platform scale and stadiometer (Scalerite, Johannesburg, South Africa). Stunting was defined as height-for-age Z-score <−2; underweight: BMI-for-age Z-score ≥−3 and <−2; overweight: BMI-for-age Z-score >1 and ≤2. Haematogram and serum ferritin were measured on the day of blood draw using a Siemens Advia 2120i Hematology System (Siemens, Munich, Germany) and the Roche COBAS Elecsys Ferritin assay (Hoffmann-La Roche, Basel, Switzerland), respectively. All sample aliquots were stored at −70 °C at the study site until shipment to the ETH Zurich, Switzerland, for further analyses. In serum, we analysed iron status biomarkers (serum ferritin, sTfR, serum iron (SFe), total iron binding capacity (TIBC), and transferrin saturation (Tsat)), systemic inflammation biomarkers (CRP, AGP), and interleukin-6 [IL-6], and hepcidin. We used a multiplex immunoassay for analyses of ferritin, sTfR, CRP, and AGP[48]. We measured hepcidin and IL-6 using ELISA kits (DRG Instruments GmbH, Germany; Research and Diagnostic Systems, Inc., US; Hycult Biotech, The Netherlands, respectively). We measured SFe and TIBC using colorimetry, and used these to calculate Tsat. IFABP was measured using a commercially available ELISA kit (Hycult Biotech, Uden, The Netherlands). Viral loads were measured using the Roche COBAS AmpliPrep/TaqMan HIV-1 Test, v2 (Hoffmann-La Roche, Basel, Switzerland), with an input volume of 1 mL allowing detection of viral loads from 20 copies per mL. Faecal calprotectin was measured using an ELISA (Eurospital, Trieste, Italy). We adjusted serum ferritin for inflammation using the Biomarkers Reflecting Inflammation and Nutritional Determinants of Anemia (BRINDA) regression correction approach[49]. Normal ranges for transferrin saturation (%) are: age <11 years 22-39%; age ≥11 years 27-44%. Normal ranges for serum iron are: age <12 years 0.53-1.19 μg/mL; age ≥12 years 0.50-1.20 μg/mL. We defined anaemia in children aged 8 to 11 years as haemoglobin <115 g/L, and in children aged 12 years or older as haemoglobin <120 g/L[40]. In participant characteristic tables, we defined ID as inflammation-adjusted ferritin <15 μg/L[43] and/or sTfR >8.3 mg/L[48] and IDA as ID and anaemia. The presence of systemic inflammation was classified as CRP ≥ 5 mg/L or AGP > 1 g/L[48].

## Iron isotopic analyses of erythrocyte incorporation

The [58]Fe as ferrous fumarate was prepared by Dr. Paul Lohmann GmbH from [58]Fe-enriched elemental iron (99.9% isotopic enrichment; Chemgas, France). The [57]Fe as ferrous sulphate was prepared at the ETH Zurich from [57]Fe-enriched elemental iron (96.2% isotopic enrichment; Chemgas, France) by dissolution in 0.1 mol/L sulfuric acid.

In Study 1, we determined FIA from the 3 iron interventions by measuring erythrocyte incorporation of the stable iron isotopes 14 days after administration as described previously[12]. Whole blood samples were mineralized in duplicate by microwave-assisted digestion in nitric acid (TurboWave, MLS), followed by iron separation as described previously[50]. We measured iron isotope ratios using an inductively coupled plasma mass spectrometer (Neptune, Thermo Finnigan) equipped with a multi-collector system for simultaneous iron beam detection[50]. We calculated the amount of [58]Fe and [57]Fe isotopic labels in blood 14 days after the administration of the second (iron-fortified maize porridge or LNS in random order) and third (oral

iron supplement) iron intervention, based on the shift in iron-isotopic ratios and the estimated amount of iron circulating in the body. Circulating iron was calculated based on haemoglobin concentrations and blood volume. The calculations were based on the methods described by Turnlund et al[51], considering that isotopic labels were not monoisotopic. For the calculation of FIA, we assumed a 75% incorporation of the absorbed iron[52].

In Study 2, we determined iron absorption, losses and gains using iron isotope dilution[12]. In the isotope dilution approach, iron of natural composition acts as a tracer diluting an ad hoc modified isotopic signature obtained via stable isotope administration and equilibration with body iron. Initially, iron isotope tracer causes enrichment in erythrocytes, which then decreases as the iron is redistributed to other tissues following erythrocyte turnover. After a period of 8 months in children, isotopic equilibrium is achieved, with the tracer concentration becoming uniform across body compartments[12]. After this point, the concentration of the isotope tracer only decreases due to the influx of iron with natural isotopic composition, which 'dilutes' the enriched tracer. The rate of decrease in the tracer concentration in circulation, expressed as the slope of tracer concentration over time, is proportional to iron absorption. The factor describing this relationship is the slope of the logarithmical tracer concentration plotted over time ($k_{abs}$). On the other hand, a decrease in the amount of tracer in the circulation can occur only with loss of the tracer and a corresponding loss of body iron (of natural isotopic composition). Iron loss only influences the amount of body iron but not its isotopic composition. The factor describing this is the slope of the logarithmical tracer amount over time ($k_{loss}$), the fraction of total body iron lost per unit of time. The mean quantity of iron absorbed ($Fe_{abs}$) over the period of interest is then calculated as $Fe_{abs} = -k_{abs} \times Fe_{total}$ where: $Fe_{total}$ is total body iron and $k_{abs}$ is the rate of change of tracer concentration constant. Mean iron loss ($Fe_{loss}$) is calculated correspondingly and net iron balance ($Fe_{gain}$) is determined by subtracting $Fe_{loss}$ from $Fe_{abs}$[12]. These calculations are described in the Supplementary Methods).

## Sample size calculations and statistical analysis

Using G*Power Statistical Program v.3.1.3., we calculated that 40 children would be needed per group in Study 1 to detect a 30% difference in FIA between groups based on a SD of 0.228 from log-transformed FIA from our previous studies, and assuming a type I error rate of 5% and power of 80%. Anticipating 10% attrition, we aimed to enrol 45 children per group. For Study 2, the isotope dilution study, with a pooled log standard deviation of 0.01 and an expected difference in body iron losses of 0.015 mg/kg body weight/day[53], we estimated that 22 subjects would be needed for group comparison with a type I error rate of 5% and 90% power. Because we expected that variability would be substantially higher than in previous studies in healthy subjects and anticipated a 30% attrition, we aimed to enrol 48 subjects per group. For Study 3, the sample size was calculated based on an expected 25% difference in serum ferritin at endpoint and an SD of 20 μg/L (ferritin values and SD were based on previously observed values in iron-replete children with HIV from the same study population[54]), which corresponded to a medium-to-large effect size of 0.625. Using an alpha of 95% and a power of 80%, the estimated sample size was 42 children per group. Considering an anticipated 5% attrition, we aimed to enrol 44 children per group.

All study data were captured and managed using REDCap electronic data capture tools hosted at ETH Zurich. We performed the statistical analyses using the R statistical programming environment (R version 4.0.2) and IBM SPSS (version 29.0.2.0). Values in the text and in tables are presented as means ± SDs for normally distributed data, and as medians (inter-quartile ranges [IQR]) for non-normally distributed data. When data were not normally distributed, appropriate transformation of values was performed before statistical analysis. Details of the statistical analyses are in the Supplementary Methods. We

considered *P* values < 0.05 as statistically significant. Results with *P* values between 0.05 and 0.08 were considered indicative of a trend and are reported as such. No adjustments were made for multiple comparisons, as primary and secondary outcomes were pre-specified and interpreted within a hierarchical framework.

### Study site and ethical statement

We conducted all studies at the Family Centre for Research with Ubuntu (FAMCRU) of the Department of Paediatrics and Child Health of Stellenbosch University at the Tygerberg Academic Hospital complex in Cape Town, South Africa. Ethical review committees of ETH Zurich (Study 1 and 2, EK 2018-N-40; Study 3, EK 2021-N-48) and Stellenbosch University (Study 1 and Study 2, M18/05/017 and S18/06/136; Study 3, M20/11/036) approved the study protocols for Studies 1, 2 and 3. We obtained written informed consent from caregivers and informed assent from children, and registered the studies at clinical-trials.gov as NCT03572010 (Studies 1 and 2) and NCT04931641 (Study 3). An independent Data Safety Monitoring Board, comprising two paediatricians and a nutritionist, monitored the three studies. The ART regimens of the children with HIV in the three studies are described in the Supplementary Methods and Supplementary Table 6.

### Reporting summary

Further information on research design is available in the Nature Portfolio Reporting Summary linked to this article.

## Data availability

The data that support the findings of this study are available from the corresponding author upon reasonable request and upon approval from the local ethics committee. The data are not publicly available due to ethical restrictions related to participant confidentiality and consent. Additional summary data and the study protocols are provided in the Supplementary Information.

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

## Acknowledgements

We thank the study participants and their caregivers; the clinical, laboratory and support staff from FAMCRU, the Division of Human Nutrition at Stellenbosch University, and the Infectious Diseases Clinic at Tygerberg Hospital; C Brand and N Robertson (Stellenbosch University, South Africa) for study coordination and data collection; C Zeder, A Groppo, T Christ, A Krzystek, K Mallick and S Menzi (ETH Zurich, Switzerland) and J Erhardt (Willstaett, Germany) for laboratory analyses; G Brittenham (Columbia University) and G Pironaci and N Stoffel (University of Oxford) for assistance with data and statistical analyses. This project was funded by the Thrasher Research Fund, USA (grant number: 14199). The funder had no role in the design of the study; in the collection, analyses, or interpretation of data; writing of the manuscript, or in the decision to publish. C.G. was also supported by the L'Oréal-UNESCO for Women in Science sub-Saharan Africa Fellowship Programme, the Harry Crossley Foundation, and the Ernst and Ethel Eriksen Trust.

## Author contributions

M.B.Z., S.L.B., M.F.C., R.B., and J.B. designed the studies. N.M., C.G., J.B., S.L.B. and R.B. conducted the studies. N.M., C.G., J.B., R.B and M.B.Z. analysed the data. M.B.Z., R.B. and J.B. wrote the first draft. M.B.Z. had primary responsibility for the final content. All authors contributed to the critical revision of the manuscript and approved the final version.

## Competing interests

The authors declare no competing interests.

## Additional information

Jeannine Baumgartner[1], Renée Blaauw [2], Nadja Mikulic[3], Charlene Goosen[2], Shaun L. Barnabas[4], Mark F. Cotton[4] & Michael B. Zimmermann [5] ✉

[1]Department of Nutritional Sciences, Faculty of Life Sciences & Medicine, King's College London, London, UK. [2]Division of Human Nutrition, Department of Global Health, Faculty of Medicine and Health Sciences, Stellenbosch University, Cape Town, South Africa. [3]Laboratory of Human Nutrition, Institute of Food, Nutrition and Health, ETH Zurich, Switzerland. [4]Family Centre for Research with Ubuntu (FAMCRU), Department of Paediatrics and Child Health, Stellenbosch University, Cape Town, South Africa. [5]MRC Translational Immune Discovery Unit, MRC Weatherall Institute of Molecular Medicine, University of Oxford, John Radcliffe Hospital, Oxford, UK. ✉e-mail: michael.zimmermann@rdm.ox.ac.uk

