## [Peer Review file · Nature Communications]

Iron absorption and loss, and efficacy of iron supplementation with and without prebiotics in children with virally suppressed HIV: three prospective studies in South Africa

Corresponding Author: Professor Michael Zimmermann

Version 0:

Reviewer comments:

Reviewer #1

(Remarks to the Author)

This manuscript reports the results of three separate but related studies of iron homeostasis (absorption and losses) in young children with and without HIV. Each of the studies applies a different method, two utilizing stable isotopes and one an RCT giving iron with or without a prebiotic (GOS), with serum ferritin as primary outcome. Substantial concerns are noted for study design. While the methodologies are each established approaches, each has limitations and the results are not equally comparable, especially for the stable isotope studies, e.g., comparing absorption for 3 different iron vehicles as single "meal"/dose (some of which are non-physiologic) for short term studies, compared to chronic absorption by different method over 6 months, where background diet and fluctuations in clinical status (inflammation) would exert more influence. Study 3 addresses different questions and arguably not would be better reported separately, which would allow more description of the 2 isotope studies and less supplementary material. The major concern is the authors' interpretation of the results.

Abstract: Please indicate age of children for each study; consider presenting dose for Study 3 relative to body weight; also suggest modify based on comments below.

Methods:

Study 1 – screening for iron status used serum Ferritin (SF) (< 40) for both HIV+ and HIV(-), is questionable; cut-off for HIV(-) should be lower (< 15 per WHO, 2020); please clarify inclusion SF criteria (lines 68-70) and basis for considering these as low SF (e.g. line 268).

Isotope studies: doses of isotope label for the maize porridge and the LNS are 14- and 3-fold greater than the intrinsic iron in the test "foods", thus providing a non-physiologic dose that would not be acting as a "tracer", and given with a single food, could not be considered to represent %iron absorption for the rest of the foods in the diet. It is not specified how the rest of the day's intake was monitored on the study days. For FeS04 tablet, the dose of isotope was more physiologic (~ 10%) but, again, absorption would reflect just that for the tablet, not rest of day. These considerations are important for comparisons to Study 2 & interpretation regarding effects of HIV, iron and inflammatory status on iron absorption.

Study 2: Inclusion criteria, again, are confusing: line 94 indicates SF ≥ 40 ug/dL (vs ug/L) but line 95 indicates same criteria as Study 1. Please clarify. Again, interpretation of iron status by SF of children w/ and w/o HIV would differ markedly. The apparent assumption that absorption of 12 mg divided into 2 doses (Study 1) is equivalent to absorption from a single dose of 12 mg is questionable.

Study 3: use of SF as primary outcome is risky, given SF's profound sensitivity to inflammation in population (HIV+) with inherent risk of inflammation. Inclusion criteria (lines 121-124) are again confusing, defining iron deficiency as SF < 30 ug/L (or sTfR > 8) &/or microcytic anemia, w/ MCV < 91.5, which would not typically be defined as microcytic. Is a different cut-off used for HIV+?

Results:

Study 1 – p 10, lower hepcidin in HIV+ face of higher inflammation is inconsistent with biology of hepcidin. Figure 3 legend or in the plots should indicate statistical comparison results – or indicate no significant differences by HIV status for any of the 3 test doses.

Discussion

Line 268 – as noted above, the criteria of SF < 40 is not considered low (if the authors are using a different analytical method

or reference range from usual (WHO, CDC, other), this should be stated; otherwise, while the groups may have had on average low SF, the high SF cut-off would promote more variability/heterogeneity within the groups and thus contributed to their responses (in absorption and supplementation)

P 14, lines 266-273 – the explanation for evidence of higher inflammation w/ lower hepcidin in children with HIV – that iron deficiency (& hypoxia-based on?) “overrode inflammatory stimulus” is contrary to observations in children with malaria and co-existent iron deficiency; other possible explanations should be considered.

P 15, 1st para (continued from prev page), might also note that this method ultimately relies on a balance study approach – subtracting one large number from another, aiming to detect relatively small differences. This is an inherent limitation/tradeoff for the outcome of interest.

P 16 – suggest note in limitations or conclusions that caution is warranted for extrapolation of absorption studies based on single “meal”/food that capture a small window of iron homeostasis. In this case, one might conclude absorption is ~ equivalent between HIV+ & HIV- children; the results of study 2 counter this. The limitations of the study 1 approach are especially relevant to a micronutrient such as iron for which homeostasis is critically controlled by absorption, which is affected by multiple factors.

Reviewer #2

(Remarks to the Author)

This manuscript presents data from three prospective studies using stable isotope techniques to investigate iron absorption and supplementation in children with virally suppressed HIV. The methods are rigorous, and the findings are clinically important. However, improvements in clarity, data presentation, and interpretation of findings may help.

Abstract

1. Line 8. Not sure why the dates are in the Findings, and not the Methods.
2. Line 12. What is the unit of “iron gains”?
3. Line 14 and 15. Where are you drawing the line for statistical significance? Good to be consistent.

Methods

4. The methodology section could benefit from a concise overview of the three studies in the beginning to orient the reader to the unique features of each.
5. I suggest including a brief rationale for the chosen dosages and formulations of iron and prebiotics.
6. Lines 204: Could the exclusion of menstruating girls from Study 2 impact generalizability of iron loss estimates?
7. Is adjustment for multiple testing necessary?

Results Interpretation

8. Again, where are you drawing the line for statistical significance? Good to be consistent, especially when p-values are in the borderline range (e.g., $P = 0.041, 0.053, 0.058, 0.088$).

Minor

9. Abbreviations should be defined consistently on first use in the main text and in figure/table legends. Check line 144 for like CRP and AGP.
10. Lines 263–265: Consider rephrasing “our findings suggest that they will likely be effective” to “our findings suggest that these strategies may be effective, particularly in virally-suppressed children on ART.”
11. Line 319: Add a forward-looking statement suggesting the need for validation of these findings in children not on ART or in malaria-endemic regions.
12. Figure 1 (line 391): Label to indicate Figure 1 applies to the selection of participants into studies 1 and 2

Reviewer #3

(Remarks to the Author)

This review concerns:

Baumgartner et al. Measurement of iron absorption and loss and the effect of iron supplementation with and without 2 prebiotics in children with virally-suppressed HIV: findings from three prospective studies in South 3 Africa.

Reviewer: Anna Chmielewska, MD, PhD, Dept.of Clinical Sciences, Pediatrics; Umeå university, Sweden

In this paper, the authors report the results of two observational studies on iron absorption and elimination, and one RCT on iron treatment with or without prebiotic (GOS) in children with virally suppressed HIV infection in South Africa. The main findings are that children with HIV have similar iron absorption from fortifiers and supplements (but higher iron absorption from foods), and similar intestinal losses, compared to children without HIV. This, in general, confirms the existing guidelines on iron supplementation in general population can be applied even in HIV positive children (at least those on antiretroviral treatment).

The RCT reported in the paper shown that adding GOS to iron fumarate tends to improve efficacy (increase of serum ferritin concentration) and safety in children with virally suppressed HIV infection and iron deficiency.

This work is of high relevance due to:

- large population that may benefit from the results (prevalence of HIV infection in children < 15 in South Africa is approx. 3%, and 6500 new vertically transmitted infections occur each year)
- high prevalence of anaemia in HIV positive children
- the complex nature of anaemia in this population, including nutritional, inflammatory and HIV-specific factors, which makes applying the knowledge on absorption and effectiveness of iron treatment from general population uncertain
- evidence within the topic is scarce and needs being reproduced
- no specific guidelines on iron treatment for HIV positive children and those for general population are being followed

The results are not completely novel and they confirm the existing evidence delivered by the same research group, showing that iron treatment is in general safe and effective in children with suppressed HIV infection.

Presenting the results of all three studies in one paper may be excessive, even though the results of the RCT are relevant and somewhat complementary to the absorption studies. Have you considered presenting the RCT results in a separate paper?

Please, see my specific comments listed below.

ABSTRACT:

- abstract gives a brief but complete picture of methodology and findings.
- Findings: please, rephrase "without significant difference in gut..." since it suggests that all other outcomes reported in the abstract are significant which they are not.
- Interpretation: concluding that prebiotic may improve efficacy based on a trend ($p=0.053$) is too strong of a statement. In general, using a term "trend" may be misleading and cause overinterpretation of the results that are not statistically significant, and so even less significant clinically (better iron efficacy in this case). The practice of considering p values <0.1 as a statistical trend has been discouraged in the research community. Even if the Authors persist to report it this way and the Editor accepts it, I suggest rephrasing the conclusion on GOS addition from "may improve" to "has shown a weak statistical trend of improving" or similar wording that will not suggest greater effect than it was.

INTRODUCTION

Line 25-26: please, refer to data of prevalence in children if available

Line 32: "using isotopic techniques" but has it been measured otherwise? If so, give a short description and describe why would your method contribute in a better way

Line 38: "would inform guidance" awkward wording

METHODS

Line 68: why s-ferritin < 40 was the inclusion criteria in a population described as iron deplete? A reader used to the s-ferritin < 15 as cut-off at this age (+ elevated CRP as exclusion) will wonder even if they will understand that it has to do with low-grade inflammation and other challenges with interpretation specific for HIV. I suggest adding a short explanation for or at least a reference.

Line 102: delete "on" before "additional 6 mo"

Line 121: how is the age of inclusion 10-15 ys motivated?

Line 123: please, explain the cut-off for MCV which is higher than the usually used

RESULTS

Congratulations to compliance rates!

Line 241: $p=0.052$ in the text, while 0.0053 in Table 3; I have not checked for more minor discrepancies

DISCUSSION

Well written discussion.

Line 258 & 311: please name that the difference was not significant

Line 323: please, rephrase so it reads better (grammar)

REFERENCES

Relevant and updated choice of cited papers. Quite a few self-citations but all are well motivated and relevant.

Version 1:

Reviewer comments:

Reviewer #1

(Remarks to the Author)

Thank you to the authors for their thoughtful and thorough responses to all of the reviews. The ms is much more cohesive with these revisions.

Reviewer #2

(Remarks to the Author)

Thanks for the opportunity to review this manuscript again. The manuscript is greatly improved and the authors have addressed the comments I made earlier. Here are some further suggestions.

1. Abstract. The phrasing "33% greater increase in iron stores" might be interpreted as definitive or causal. I suggest using

the phrase "33% lower serum ferritin" instead.

2. Discussion, Study 2. The magnitude of the difference in iron absorption in children with HIV vs counterparts without was small: 1.09 vs 1.44 mg/d. Do you agree? If yes, I suggest acknowledging that the clinical impact of that difference may be minimal, especially in populations with high dietary diversity.

Point-by-Point Response to Reviewers' Comments

Manuscript NCOMS-25-20615: "Measurement of iron absorption and loss and iron supplementation with prebiotics in children with HIV"

Response to Reviewer #1

This manuscript reports the results of three separate but related studies of iron homeostasis (absorption and losses) in young children with and without HIV. Each of the studies applies a different method, two utilizing stable isotopes and one an RCT giving iron with or without a prebiotic (GOS), with serum ferritin as primary outcome. Substantial concerns are noted for study design. While the methodologies are each established approaches, each has limitations and the results are not equally comparable, especially for the stable isotope studies, e.g., comparing absorption for 3 different iron vehicles as single "meal"/dose (some of which are non-physiologic) for short term studies, compared to chronic absorption by different method over 6 months, where background diet and fluctuations in clinical status (inflammation) would exert more influence. Study 3 addresses different questions and arguably not would be better reported separately, which would allow more description of the 2 isotope studies and less supplementary material. The major concern is the authors' interpretation of the results.

Our response:

We thank the reviewer for their thoughtful and detailed comments. We agree that the three studies differ in design, methodology, and duration, and that each has inherent limitations. However, we respectfully believe that presenting them together provides a more comprehensive and integrated understanding of iron metabolism in children with HIV, a population for whom data remain limited, and therefore is more likely to inform future policy and practice.

While the methods differ, the studies were conducted in overlapping populations, within the same geographic and clinical context, and were conceptually linked by the overarching aim of informing iron intervention strategies in children with HIV. We have revised the manuscript to more clearly delineate the scope and limitations of each study, specifically in the Introduction and Discussion (see revised Strengths and Limitations; lines 246-264). The comment related to "dose" is answered below.

Abstract: Please indicate age of children for each study; consider presenting dose for Study 3 relative to body weight; also suggest modify based on comments below.

Our response:

In revising the abstract, we followed Nature Communications formatting guidelines, which require a concise, non-technical summary of approximately 150 words. We were therefore unable to include detailed methodological information, such as age ranges and dosing specifics, within the abstract itself. However, in the revised manuscript we ensured that the age ranges of participants are clearly stated in the Results section for each study, which is now presented before the methods section and therefore contains a brief summary of each study design.

Methods:

Study 1 – screening for iron status used serum Ferritin (SF) (< 40) for both HIV+ and HIV(-), is questionable; cut-off for HIV(-) should be lower (< 15 per WHO, 2020); please clarify inclusion SF criteria (lines 68-70) and basis for considering these as low SF (e.g. line 268).

Our response:

According to the 2020 WHO Guideline on the use of ferritin concentrations to assess iron status, a ferritin threshold $<15 \mu\text{g/L}$ is recommended to define iron deficiency in apparently healthy children without infection or inflammation. However, in Study 1, approximately half of the participants were children living with HIV—a population known to have elevated levels of inflammation. Using the standard threshold of $<15 \mu\text{g/L}$ without accounting for inflammation (as measured by CRP and AGP) would likely have underestimated iron deficiency in this group. Since it was not feasible to measure CRP and AGP at screening, we adopted an alternative approach by raising the ferritin threshold used for inclusion to better account for potential inflammation-related bias. Current WHO guidelines recommend a ferritin threshold of $<70 \mu\text{g/L}$ to define iron deficiency in children aged ≥ 5 years with inflammation. As a compromise, we set the inclusion threshold at $<40 \mu\text{g/L}$, aiming to capture children with depleted iron stores while minimizing misclassification. This threshold is supported by recent studies in infants and women (PMID: 36811475; PMID: 344016875), which demonstrate that early iron deficiency—as indicated by increased iron absorption, one of the earliest physiological responses to low iron status—occurs at ferritin concentrations around $45 \mu\text{g/L}$. Furthermore, a recent technical review also supports the use of a ferritin cut-off around $40 \mu\text{g/L}$ to define iron deficiency (PMC10824166).

We added the following justification to the Methods section:

Lines 295-300: *“For Study 1, we included children with depleted iron stores by applying a serum ferritin threshold of $\leq 40 \mu\text{g/L}$, based on recent evidence indicating that iron absorption begins to increase at ferritin concentrations around $45 \mu\text{g/L}$ in both infants and women of reproductive age—a physiological response to early iron deficiency. Also, most recent WHO guidelines recommend a ferritin cut-off of $<70 \mu\text{g/L}$ for defining iron deficiency in children ≥ 5 years with inflammation, further supporting the use of a higher threshold.”*

In the Result section (participant characteristics), we reported the prevalence of iron deficiency defined as inflammation-adjusted ferritin $<15 \mu\text{g/L}$.

We have, however, changed the wording in our manuscript to refer to the children included in Study 1 as iron-depleted rather than iron-deficient.

Isotope studies: doses of isotope label for the maize porridge and the LNS are 14- and 3-fold greater than the intrinsic iron in the test “foods”, thus providing a non-physiologic dose that would not be acting as a “tracer”, and given with a single food, could not be considered to represent %iron absorption for the rest of the foods in the diet. It is not specified how the rest of the day’s intake was monitored on the study days. For FeS04 tablet, the dose of isotope was more physiologic ($\sim 10\%$) but, again, absorption would reflect just that for the tablet, not rest of day. These considerations are important for comparisons to Study 2 & interpretation regarding effects of HIV, iron and inflammatory status on iron absorption.

Our response:

We agree that the isotope doses used in Study 1 were substantially higher than the intrinsic iron content of the maize porridge and LNS. However, our intention was not to assess the absorption of native dietary iron, but rather to evaluate absorption of added (extrinsic) iron, as typically provided through public health fortification of foods (e.g., iron-fortified maize meal or wheat flour in South Africa) or supplementation programmes (e.g., iron-containing LNS or oral iron tablets). The isotope-

labelled iron was added in amounts that reflect commonly used fortification or supplementation doses. The test meals were therefore designed to model real-world iron interventions rather than habitual dietary iron intake.

As noted, the design of Study 1 was intended to measure fractional iron absorption from a single dose or meal and does not capture the cumulative effects of long-term intake alongside habitual dietary iron. We have revised the manuscript to more clearly articulate the rationale for the different studies in the revised manuscript. See revised introduction and discussion.

Study 2: Inclusion criteria, again, are confusing: line 94 indicates SF ≥ 40 $\mu\text{g}/\text{dL}$ (vs $\mu\text{g}/\text{L}$) but line 95 indicates same criteria as Study 1. Please clarify. Again, interpretation of iron status by SF of children w/ and w/o HIV would differ markedly. The apparent assumption that absorption of 12 mg divided into 2 doses (Study 1) is equivalent to absorption from a single dose of 12 mg is questionable.

Our response:

We agree that the description of the inclusion criteria for Study 2 were confusing. We included children who had previously participated in Study 1 and recruited additional participants using the same inclusion and exclusion criteria, with one exception: children with serum ferritin (SF) ≥ 40 $\mu\text{g}/\text{L}$ were also eligible for inclusion (i.e., there was no ferritin threshold in Study 2). The use of $\mu\text{g}/\text{dL}$ was an error and has been corrected.

Regarding the 12 mg iron dose: equivalence of absorption between a single 12 mg dose and two smaller doses is not relevant in this context, as iron absorption from this dose was not assessed. The purpose of administering the 12 mg ^{57}Fe -labelled dose was to enrich all body iron compartments with the stable isotope over the 8-month equilibration period in preparation for Study 2.

Study 3: use of SF as primary outcome is risky, given SF's profound sensitivity to inflammation in population (HIV+) with inherent risk of inflammation. Inclusion criteria (lines 121-124) are again confusing, defining iron deficiency as SF < 30 $\mu\text{g}/\text{L}$ (or sTfR > 8) &/or microcytic anemia, w/ MCV < 91.5 , which would not typically be defined as microcytic. Is a different cut-off used for HIV+?

Our response:

We agree that the use of serum ferritin (SF) as a primary outcome has limitations due to its sensitivity to inflammation. However, it is the recommended iron biomarker to assess iron interventions (Lynch S, et al. J Nutr. 2018;148(suppl1):1001S-1067S.) Given that this study was conducted exclusively in children with HIV, and there were relatively few intercurrent infections during the study period, we considered SF to remain a sensitive and clinically relevant marker for detecting changes in iron status in response to the adjunct treatment with GOS over 12 weeks. Moreover, we applied the BRINDA correction method to adjust ferritin values for inflammation.

For transparency, we also report the prevalence of iron deficiency using a more conservative SF threshold of < 15 $\mu\text{g}/\text{L}$ (using inflammation-adjusted ferritin values) in the Results section.

Thank you for picking up the issue with the MCV cut-off. We intended to include children with normocytic or microcytic anaemia. A cut-off ≥ 91.5 fl is used by the National Health Laboratory Service of South Africa to detect macrocytic anaemia. We revised the sentence accordingly:

Lines 371-374: "(iv) iron-deficient, defined as serum ferritin < 30 $\mu\text{g}/\text{L}$ or sTfR > 8.3 mg/L , and/or normocytic or microcytic anaemic, defined as haemoglobin $< 110/120$ (children 10–11/12–15 years) g/L and mean corpuscular volume (MCV) < 91.5 fL (cut-off value used by the National Health Laboratory Service of South Africa)."

Results:

Study 1 – p 10, lower hepcidin in HIV+face of higher inflammation is inconsistent with biology of hepcidin. Figure 3 legend or in the plots should indicate statistical comparison results – or indicate no significant differences by HIV status for any of the 3 test doses.

Our response:

We were also surprised by the relatively low hepcidin levels in the children with HIV. However, we are confident that analyses were conducted correctly. See our explanation in the Discussion section and our response to the related comment below.

Thank you, we have now revised the legend of Figure 3.

Discussion

Line 268 – as noted above, the criteria of SF < 40 is not considered low (if the authors are using a different analytical method or reference range from usual (WHO, CDC, other), this should be stated; otherwise, while the groups may have had on average low SF, the high SF cut-off would promote more variability/heterogeneity within the groups and thus contributed to their responses (in absorption and supplementation)

Our response:

A SF <40 ug/L was used solely as an inclusion criterion for Study 1, as the objective was to compare iron absorption from different iron interventions in children with depleted iron stores—not necessarily those who were clinically iron deficient. Across all studies, we also reported iron status using the standard <15 µg/L ferritin cut-off, adjusted for inflammation. As noted, the three studies were designed to address different aspects of iron intervention strategies for both the prevention and treatment of iron deficiency in children with virally suppressed HIV. We therefore believe that the heterogeneity in iron status across the studies enhances the generalisability of the findings to the broader population of children with HIV.

P 14, lines 266-273 – the explanation for evidence of higher inflammation w/ lower hepcidin in children with HIV – that iron deficiency (& hypoxia-based on?) “overrode inflammatory stimulus” is contrary to observations in children with malaria and co-existent iron deficiency; other possible explanations should be considered.

Our response:

We thank the reviewer for this insightful comment. We agree that the relationship between inflammation, iron deficiency, and hepcidin regulation is complex and highly context-dependent. While our original interpretation was based on evidence of hepcidin suppression in iron-deficient states—such as in adult women without HIV—we acknowledge that in certain conditions, including acute infections like malaria and tuberculosis, inflammatory signals may predominate and lead to elevated hepcidin levels despite concurrent iron deficiency. We have revised the discussion to reflect this nuance. Specifically, we have added:

Lines 202-203: “However, in other contexts, such as acute infections like malaria and tuberculosis, co-existing inflammation may dominate and elevate hepcidin despite underlying iron deficiency.”

We also replaced “hypoxia” with “erythropoiesis”, on the basis that children with HIV had higher sTfR concentrations and borderline lower anaemia.

P 15, 1st para (continued from prev page), might also note that this method ultimately relies on a balance study approach – subtracting one large number from another, aiming to detect relatively small differences. This is an inherent limitation/tradeoff for the outcome of interest.

Our response:

We are not entirely clear which outcome the comment refers to, but we assume it pertains to the isotope dilution method used in Study 2 to estimate long-term iron absorption and losses. This approach differs from a traditional balance study, which typically involves measuring total dietary intake and excretion. Instead, the isotope dilution method estimates iron absorption and loss based on the dilution of an administered stable isotope within the body's iron pool over time by iron of natural isotopic composition (from diet, fortified foods, or supplements). Therefore, importantly, this method does not depend on subtracting two large values to detect small differences.

P 16 – suggest note in limitations or conclusions that caution is warranted for extrapolation of absorption studies based on single “meal”/food that capture a small window of iron homeostasis. In this case, one might conclude absorption is ~ equivalent between HIV+ & HIV- children; the results of study 2 counter this. The limitations of the study 1 approach are especially relevant to a micronutrient such as iron for which homeostasis is critically controlled by absorption, which is affected by multiple factors.

Our response:

We thank the reviewer for this important comment. We agree that single-dose iron absorption studies provide only a limited assessment of iron homeostasis and may not fully reflect habitual absorption patterns. In response, we have revised the limitations section of the Discussion to explicitly address this. The revised text now reads:

Lines 256-259: *“Second, in Study 1, iron absorption was assessed following a single administration of each intervention under fasting conditions. Such single-dose studies may not fully represent habitual absorption patterns and may overestimate or underestimate an individual’s true iron absorption capacity.”*

Response to Reviewer #2

This manuscript presents data from three prospective studies using stable isotope techniques to investigate iron absorption and supplementation in children with virally suppressed HIV. The methods are rigorous, and the findings are clinically important. However, improvements in clarity, data presentation, and interpretation of findings may help.

Our response:

Thank you for the positive feedback regarding methodological rigour and clinical importance. We appreciate your suggestions to improve clarity, data presentation, and interpretation. We have revised the manuscript accordingly and believe these changes have strengthened the overall quality of the work.

Abstract

Our response:

Thank you for your comments on the abstract. In revising the abstract, we followed Nature Communications formatting guidelines, which require a concise, non-technical summary of approximately 150 words. We were therefore unable to include detailed methodological information.

1. Line 8. Not sure why the dates are in the Findings, and not the Methods.

Our response:

Removed in revised abstract.

2. Line 12. What is the unit of “iron gains”?

Our response:

Detail not presented in revised non-technical abstract.

3. Line 14 and 15. Where are you drawing the line for statistical significance? Good to be consistent.

Our response:

Thank you for highlighting the need for consistency in reporting statistical significance. We have revised the abstract to remove the reference to gut integrity ($P=0.088$) and now report only outcomes with $P < 0.08$ as indicative of trends. We also defined this threshold in the Statistical Analysis section for clarity and consistency.

Line 480-481: *“Results with P values between 0.05 and 0.08 were considered indicative of a trend and are reported as such.”*

Methods

4. The methodology section could benefit from a concise overview of the three studies in the beginning to orient the reader to the unique features of each.

Our response:

We completely re-formatted the manuscript to align with Nature Communication guidelines. Therefore, the methods section was moved to the end of the manuscript (after the Discussion) and a concise summary of the design has been added to the results section for each study. We have also provided a more detailed rationale for each of the three studies. We believe this has improved clarity.

5. I suggest including a brief rationale for the chosen dosages and formulations of iron and prebiotics.

Our response:

Thank you for this valuable suggestion. We have added the following statement to the results section of Study 1:

Lines 66-69: *“Between September 2018 and September 2019, we conducted a prospective, cross-over study with case-control comparisons to assess iron absorption from three commonly used iron interventions, at doses representative of public health and clinical practice, in iron-depleted children with and without HIV.”*

And to the results section of Study 3:

Lines 150-153: *“Participants received 50 mg of elemental iron as ferrous fumarate, consistent with the South African Standard Treatment Guidelines and Essential Medicines List. The GOS dose (7.5 g) was*

selected based on prior evidence demonstrating its potential to enhance iron absorption from ferrous fumarate and mitigating adverse effects of iron fortification on the gut microbiome."

6. Lines 204: Could the exclusion of menstruating girls from Study 2 impact generalizability of iron loss estimates?

Our response:

Thank you for raising this important issue. We have now performed a sensitivity analysis, including the 15 menstruating girls in the analyses and reported the findings in the results section:

Lines 130-135: *"In a sensitivity analysis that included the 15 menstruating girls, median (IQR) daily iron absorption (Feabs) was not significantly different between children with HIV (n=35) and those without HIV (n=45): 1.10 (0.88–1.47) mg/day vs. 1.42 (0.95–1.81) mg/day, respectively (P = 0.119). However, median (IQR) iron loss (Feloss) was significantly lower in children with HIV: 0.90 (0.46–1.16) mg/day compared to 0.97 (0.61–1.77) mg/day in children without HIV (P = 0.036). Net iron balance (Fegain) was similar between groups: 0.29 (–0.05 to 0.72) mg/day in children with HIV and 0.32 (–0.29 to 0.84) mg/day in those without HIV (P = 0.372)."*

7. Is adjustment for multiple testing necessary?

Our response:

We believe that adjustment for multiple testing is not necessary in this context, as our study was designed with clearly defined a priori primary and secondary outcomes. The primary outcome (serum ferritin) was specified in advance and forms the basis of our main hypothesis testing. Secondary outcomes were also pre-specified and are interpreted in a supportive and exploratory context, rather than as independent confirmatory tests.

We added the following statement to the statistical methods section:

Lines 481-482: *"No adjustments were made for multiple comparisons, as primary and secondary outcomes were pre-specified and interpreted within a hierarchical framework."*

Results Interpretation

8. Again, where are you drawing the line for statistical significance? Good to be consistent, especially when p-values are in the borderline range (e.g., P = 0.041, 0.053, 0.058, 0.088).

Our response:

We agree that consistency in defining statistical significance is important, particularly for borderline P values. We have now clearly defined statistical significance as $P < 0.05$ and report results with P values between 0.05 and 0.08 as indicative of trends. This threshold is now specified in the Statistical Analysis section, and we have revised the text accordingly to ensure consistent interpretation throughout the manuscript.

Line 480-481: *"We considered P values <0.05 as statistically significant. Results with P values between 0.05 and 0.08 were considered indicative of a trend and are reported as such."*

Minor

9. Abbreviations should be defined consistently on first use in the main text and in figure/table legends. Check line 144 for like CRP and AGP.

Our response: Thank you, we have checked all our abbreviations and made sure they are defined consistently on first use in the main text and in figure/table legends

10. Lines 263–265: Consider rephrasing “our findings suggest that they will likely be effective” to “our findings suggest that these strategies may be effective, particularly in virally-suppressed children on ART.”

Our response: Thank you for this suggestion. See earlier revised statement in the Discussion:

Lines 192-193: *“Our findings suggest that these strategies may still be effective, particularly in virally suppressed children receiving ART.”*

11. Line 319: Add a forward-looking statement suggesting the need for validation of these findings in children not on ART or in malaria-endemic regions.

Our response: Thank you. We have now incorporated such a statement in the discussion.

Line 242-245: *“If confirmed by larger studies and validated in children not receiving ART or living in malaria-endemic regions, these findings suggest prebiotics may improve both the efficacy and safety of oral iron supplementation in iron-deficient children with HIV.”*

12. Figure 1 (line 391): Label to indicate Figure 1 applies to the selection of participants into studies 1 and 2.

Our response: Thank you, we have revised the legends for Figures 1 and 2 accordingly.

Response to Reviewer #3

In this paper, the authors report the results of two observational studies on iron absorption and elimination, and one RCT on iron treatment with or without prebiotic (GOS) in children with virally suppressed HIV infection in South Africa.

The main findings are that children with HIV have similar iron absorption from fortifiers and supplements (but higher iron absorption from foods), and similar intestinal losses, compared to children without HIV. This, in general, confirms the existing guidelines on iron supplementation in general population can be applied even in HIV positive children (at least those on antiretroviral treatment).

The RCT reported in the paper shown that adding GOS to iron fumarate tends to improve efficacy (increase of serum ferritin concentration) and safety in children with virally suppressed HIV infection and iron deficiency.

This work is of high relevance due to:

- large population that may benefit from the results (prevalence of HIV infection in children < 15 in South Africa is approx. 3%, and 6500 new vertically transmitted infections occur each year)
- high prevalence of anaemia in HIV positive children
- the complex nature of anaemia in this population, including nutritional, inflammatory and HIV-specific factors, which makes applying the knowledge on absorption and effectiveness of iron treatment from general population uncertain
- evidence within the topic is scarce and needs being reproduced

- no specific guidelines on iron treatment for HIV positive children and those for general population are being followed

The results are not completely novel and they confirm the existing evidence delivered by the same research group, showing that iron treatment is in general safe and effective in children with suppressed HIV infection.

Presenting the results of all three studies in one paper may be excessive, even though the results of the RCT are relevant and somewhat complementary to the absorption studies. Have you considered presenting the RCT results in a separate paper?

Our response: We sincerely thank Reviewer #3 for their thoughtful and detailed evaluation of our manuscript. We appreciate your recognition of the relevance and potential impact of our findings, particularly in the context of the high burden of HIV and anaemia in children in South Africa and beyond. Your summary of our study objectives and key findings is accurate (except that iron absorption from habitual diet (foods) was lower in children with compared to without HIV), and we are grateful for your acknowledgment of the contribution this work makes to a relatively under-researched area.

We also appreciate your suggestion regarding the inclusion of the RCT results in this manuscript. While we considered separating the RCT into a standalone manuscript, we felt that integrating the findings from all three studies provided a more comprehensive understanding of iron absorption and treatment in this population. However, we have revised the manuscript to ensure that the structure is clear and that the distinct contributions of each study are well delineated.

Please, see my specific comments listed below.

ABSTRACT:

- abstract gives a brief but complete picture of methodology and findings.

Our response:

Thank you. Just to note that in revising the abstract, we now had to follow Nature Communications formatting guidelines, which require a concise, non-technical summary of approximately 150 words. As such, we were unable to include detailed methodological information.

- Findings: please, rephrase “without significant difference in gut...” since it suggests that all other outcomes reported in the abstract are significant which they are not.

Our response:

See above regarding the revising of the abstract, which resolved this issue.

- Interpretation: concluding that prebiotic may improve efficacy based on a trend ($p=0.053$) is too strong of a statement.

In general, using a term “trend” may be misleading and cause overinterpretation of the results that are not statistically significant, and so even less significant clinically (better iron efficacy in this case). The practice of considering p values <0.1 as a statistical trend has been discouraged in the research community. Even if the Authors persist to report it this way and the Editor accepts it, I suggest rephrasing the conclusion on GOS addition from “may improve” to “has shown a weak statistical trend of improving” or similar wording that will not suggest greater effect than it was.

Our response:

We agree that the interpretation of findings based on borderline P values should be cautious to avoid overstatement. As recommended by Reviewer #2, we have now clearly defined statistical significance as $P < 0.05$ and report results with P values between 0.05 and 0.08 as indicative of trends (statement added to Statistical Analysis section).

We have now clearly presented this finding as a trend in the manuscript.

INTRODUCTION

Line 25-26: please, refer to data of prevalence in children if available

Our response:

Thank you, we have added data on the prevalence of anaemia in children with HIV.

Lines 15-17: *"The global prevalence of anaemia among children under 15 years with HIV is ~40%, with reported rates ranging from 26% to 61%, depending on setting and disease stage."*

Line 32: "using isotopic techniques" but has it been measured otherwise? If so, give a short description and describe why would your method contribute in a better way

Our response:

Thank you for this helpful comment. We have revised the relevant section of the introduction to better highlight the contribution of our study, specifically study 1.

Lines 25-32: *"Despite these concerns, to our knowledge, iron absorption from fortified foods and supplements has not been previously measured in children with HIV using isotopic methods, nor directly compared to children without HIV."*

Previous studies investigating the response to iron interventions in children with HIV have relied on conventional iron status biomarkers, such as serum ferritin and soluble transferrin receptor (sTfR), both of which are influenced by inflammation and may not reliably reflect iron status. In contrast, stable isotope techniques provide a direct, accurate and safe method for quantifying iron absorption and losses, independent of inflammation."

Line 38: "would inform guidance" awkward wording

Our response:

Thank you. We changed the statement to (lines 61-63): *"Improved understanding of iron metabolism and treatment response in this vulnerable group is essential to inform evidence-based strategies for the prevention and management of iron deficiency in paediatric HIV."*

METHODS

Line 68: why s-ferritin < 40 was the inclusion criteria in a population described as iron deplete? A reader used to the s-ferritin < 15 as cut-off at this age (+ elevated CRP as exclusion) will wonder even if they will understand that it has to do with low-grade inflammation and other challenges with interpretation specific for HIV. I suggest adding a short explanation for or at least a reference.

Our response:

Thank you for this comment. Reviewer #1 has also questioned our choice for this threshold as inclusion criteria (see our response above). We have now added the following rationale for this inclusion threshold in Study 1:

Lines 295-300: *“For Study 1, we included children with depleted iron stores by applying a serum ferritin threshold of $\leq 40 \mu\text{g/L}$, based on recent evidence indicating that iron absorption begins to increase at ferritin concentrations around $45 \mu\text{g/L}$ in both infants and women of reproductive age—a physiological response to early iron deficiency. Also, most recent WHO guidelines recommend a ferritin cut-off of $< 70 \mu\text{g/L}$ for defining iron deficiency in children ≥ 5 years with inflammation, further supporting the use of a higher threshold.”*

In the Results section (participant characteristics), we report the prevalence of iron deficiency defined as inflammation-adjusted ferritin $< 15 \mu\text{g/L}$.

We have changed the wording in our manuscript to refer to the children included in Study 1 as iron-depleted rather than iron deficient.

Line 102: delete “on” before “additional 6 mo”

Our response:

Done, thank you.

Line 121: how is the age of inclusion 10-15 ys motivated?

Our response:

The age range of 10–15 years was selected to allow inclusion of children who had previously participated in Study 1 and/or Study 2. This age range reflected the age of eligible participants from those studies at the time Study 3 was initiated.

We added the following statement to the Methods section (lines 366-368): *“Participants were recruited from among those with HIV in Study 1 and/or 2 ($n=32$). Accordingly, the age range for inclusion in this study was set at 10–15 years to match the ages of these participants at the time of recruitment.”*

Line 123: please, explain the cut-off for MCV which is higher than the usually used

Our response:

Thank you for picking up the issue with the MCV cut-off. We intended to include children with normocytic or microcytic anaemia. A cut-off $\geq 91.5 \text{ fL}$ is used by the National Health Laboratory Service of South Africa to detect macrocytic anaemia. We revised the sentence accordingly:

Lines 371-374: *“(iv) iron-deficient, defined as serum ferritin $< 30 \mu\text{g/L}$ or sTfR $> 8.3 \text{ mg/L}$, and/or normocytic or microcytic anaemic, defined as haemoglobin $< 110/120$ (children 10–11/12–15 years) g/L and mean corpuscular volume (MCV) $< 91.5 \text{ fL}$ (cut-off value used by the National Health Laboratory Service of South Africa).”*

RESULTS

Congratulations to compliance rates!

Our response:

Thank you. We believe the high compliance rate was largely due to including children from an already established cohort. The research team had built a strong relationship of trust with both the caregivers and children over time, which likely contributed to their continued engagement and adherence.

Line 241: $p=0.052$ in the text, while 0.053 in Table 3; I have not checked for more minor discrepancies

Our response:

Thank you. We have corrected the value in the text.

DISCUSSION

Well written discussion.

Our response:

Thank you.

Line 258 & 311: please name that the difference was not significant

Our response:

Done.

Line 323: please, rephrase so it reads better (grammar)

Our response:

Rephrased.

REFERENCES

Relevant and updated choice of cited papers. Quite a few self-citations but all are well motivated and relevant.

Our response:

Thank you.

Point-by-Point Response to Reviewers' Comments

Manuscript NCOMS-25-20615A "Iron absorption and loss and efficacy of iron supplementation with and without prebiotics in children with virally suppressed HIV: three prospective studies in South Africa"

Response to Reviewer #1

Thank you to the authors for their thoughtful and thorough responses to all of the reviews. The ms is much more cohesive with these revisions.

Our response:

We thank the Reviewer for their positive feedback and are pleased that the revised manuscript is now considered more cohesive. We greatly appreciate the Reviewer's thoughtful comments, which helped us improve the clarity and overall quality of the manuscript.

Response to Reviewer #2

Thanks for the opportunity to review this manuscript again. The manuscript is greatly improved, and the authors have addressed the comments I made earlier.

Our response: We thank the reviewer for their positive feedback and are pleased to hear that we have adequately addressed the Reviewer's earlier comments.

Here are some further suggestions.

1. Abstract. The phrasing "33% greater increase in iron stores" might be interpreted as definitive or causal. I suggest using the phrase "33% lower [higher] serum ferritin" instead.

Our response: We agree and have revised the abstract to refer specifically to serum ferritin. We also identified an error in our earlier calculation: the relative increase in serum ferritin over the intervention period is 39%, not 33%. Median (IQR) serum ferritin increased from 14.9 (9.0–23.4) to 40.8 (21.5–72.3) µg/L in the iron+GOS group (+174%), and from 13.6 (9.7–21.5) to 30.6 (16.5–50.2) µg/L in the iron+placebo group (+125%), corresponding to a 39% greater relative increase in the intervention group. The abstract and relevant sections of the manuscript have been updated accordingly.

We have revised the abstract and relevant information in manuscript accordingly:

"In the third study, a 12-week randomised, placebo-controlled, double-blind trial, iron-deficient children with HIV receiving iron with prebiotic galacto-oligosaccharides (n=41) exhibit a borderline-significant (p=0.053) 39% greater relative increase in serum ferritin (primary outcome) over the intervention period compared to those receiving iron with placebo (n=42),..."

2. Discussion, Study 2. The magnitude of the difference in iron absorption in children with HIV vs counterparts without was small: 1.09 vs 1.44 mg/d. Do you agree? If yes, I suggest acknowledging that the clinical impact of that difference may be minimal, especially in populations with high dietary diversity.

Our response: We respectfully disagree with the reviewer. While the absolute difference in iron absorption (1.09 vs 1.44 mg/day) may appear modest, it represents a 24% lower relative absorption in children with HIV. Relative differences of this magnitude are typically considered clinically relevant. Moreover, the median iron absorption in children with HIV (1.09 mg/day) was below the estimated absorbed iron requirement for this age group (1.17–1.20 mg/day), as discussed in the manuscript. We have interpreted this finding with nuance, attributing the lower absorption to both biological (e.g., inflammation) and dietary factors, which are shaped by socio-economic disparities. Hence, we would agree with the Reviewer that in populations with high dietary diversity, children with virally suppressed HIV may be able to absorb adequate amounts of iron, but poor dietary diversity remains a major cause of iron deficiency in children in low- and middle-income countries.